# Quantifying QiGan: What Shape Patterns Does a DNN Exploit in Go?

## Abstract

Given a deep neural network (DNN) that has surpassed human beings in a task, disentangling the explicit knowledge encoded by the DNN to gain new insights into the task is a promising yet challenging regime in explainable AI. In this paper, we aim to disentangle the "QiGan[1]" encoded by the AI model for Go, which has beaten top human players. Specifically, we disentangle primitive shape patterns[2] of stones memorized by the value network, and these shape patterns represent the "QiGan[1]" used to conduct a fast situation assessment of the current board state. To this end, we propose to use both AND interactions[3] and OR interactions to obtain a more concise explanation of shape patterns. We further prove the universal matching property of AND-OR interactions to ensure the faithfulness and verifiability of the explained shape patterns. In experiments, our method finds many novel shape patterns beyond the traditional shape patterns in human knowledge.

## 1 Introduction

The explanation for AI models has gained increasing attention in recent years. However, in this paper, we consider a new problem, *i.e.*, if an AI model has achieved superior performance to human beings in a task, *how can we clearly disentangle exact inference patterns used by the AI model to help people better understand hidden rules of the task?*

Because AI models for the Go game have been widely regarded to have surpassed human players (Granter et al., 2017; Fang et al., 2018; Intelligence, 2016), in this paper, let us consider the Go game as a case study, and disentangle shape patterns used by AI models[4] to play the Go game.

In fact, the disentanglement of shape patterns is particularly valuable for the Go game. Because the Go game is one of the most difficult games, with a game-tree complexity of $10^{360}$, far beyond the limits of human or computer capabilities (Lee, 2004; Van Der Werf, 2004), both people and the DNN have to learn and use "QiGan[1]" as a fast intuitive situation assessment. In comparison, other games (*e.g.* chess and poker) rely primarily on search capabilities. In the AI model for the Go game, the value network encodes the "QiGan[1]", while the policy network and the Monte Carlo Tree Search correspond to the searching capability (Gelly & Silver, 2011; Buesing et al., 2020).

**What is a faithful explanation of the shape patterns encoded by a DNN?** In this paper, we aim to discover explicit new shape patterns of stones (*i.e.*, QiGan[1]) used by the value network to estimate the advantage score. Unlike mechanistic interpretability studies (Elhage et al., 2021; Gao et al., 2025) (see Appendix B), we do not explain specific neurons in a DNN. Instead, we try to disentangle the shape patterns (QiGan[1]) encoded by the *entire* value network, which are represented by interactions among stones in the game board. As Figure 1 shows, each AND interaction represents a shape pattern formed by all stones in set $S$. If and only if all stones in $S$ are present on the board, then this AND interaction is activated and contributes an effect to influence the value network's inference.

The faithfulness of taking interactions as primitive shape patterns encoded by the DNN is guaranteed by the principle of using sufficiently few interactions to precisely mimic the DNN's outputs

---

[1]In the community of the Go game, "QiGan" is widely used to refer to the fast situation assessment based on shape patterns of stones without a sophisticated search. "QiGan" can also be replaced with "knowledge."

[2]Shape patterns, refer to the various shapes formed by the arrangement of stones on the board.

[3]Please see the ***video demo*** in the supplementary material for the interaction-based explanation.

[4]We use the KataGo (Wu, 2019) for testing, because the AlphaGo is not open-sourced.

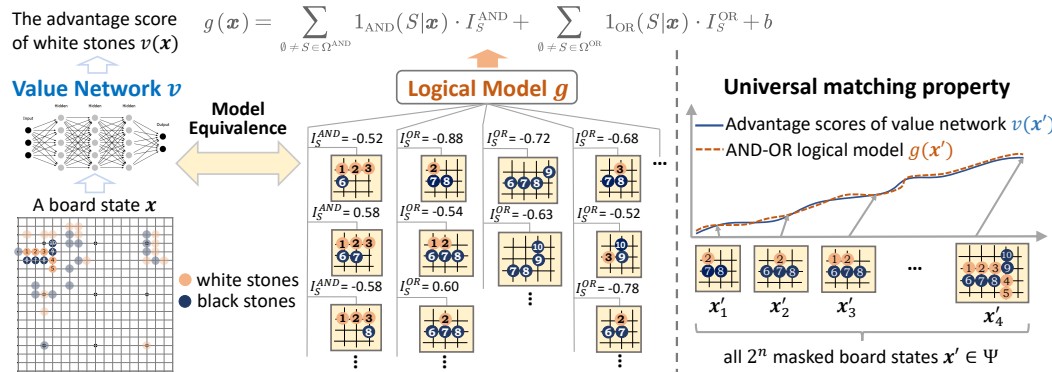

Figure 1: (Left): We use a logical model $g$ with AND-OR interactions to explain detailed shape patterns encoded by the value network $v$. Each interaction represents a shape pattern, corresponding to either an AND or OR relationship among a set $S$ of input variables (stones). (Right): The universal matching property shows that the advantage scores estimated by $v$ on all masked board states $\boldsymbol{x}' \in \Psi$ can be precisely mimicked by $g$, which guarantees the faithfulness of the explanation.

on sufficiently many board states, which can be summarized by two requirements. **(1) Fidelity requirement:** interactions must be powerful enough to accurately predict all outputs of the DNN on a large set of inputs, which is termed the *universal matching property* of interactions. **(2) Conciseness requirement:** the interactions used for explanation should be sufficiently few, which is termed the *sparsity property* of interactions.

**Developing OR interactions to explain OR relationships.** However, we discover and prove that traditional AND interactions cannot concisely represent the OR relationship encoded by the value network. Specifically, a single OR relationship among $m$ stones will be redundantly explained as exponentially many ($2^m - 1$) different AND interactions.

Therefore, we extend AND interactions to define OR interactions. More crucially, we theoretically prove and experimentally verify the faithfulness of the AND-OR interactions from three perspectives. (1) We prove that simultaneously using AND and OR interactions can accurately explain the complex inference logic of the value network as much sparser shape patterns, compared to using only AND interactions. (2) We prove that the AND-OR interactions also retain the universal matching property. (3) We find that shape patterns extracted from one board state can be transferred to explain the value network's outputs on other board states.

**Identifying common coalitions used by the value network.** Furthermore, we observe that some complex shape patterns (*i.e.*, interactions) in the Go game may contain too many stones. Such overly complex shape patterns are typically considered as unusual shape patterns memorized by the value network, rather than common shape patterns shared by different board states. Thus, we further identify recurring combinations of stones that appear in different interactions/shapes, which we refer to as *common coalitions*. For example, Figure 1 shows that both the interactions $S_1 = \{1, 2, 3, 6\}, S_2 = \{1, 2, 3, 8\}$ contain the coalition $T = \{1, 2, 3\}$. We apply the attribution method (Xinhao Zheng, 2025) to estimate the attribution of each coalition to help humans understand the DNN's logic.

We collaborate with professional human Go players[13] to compare interactions/coalitions encoded by the value network with their QiGan[1] of the Go game. We find that some identified shape patterns align with human understanding, while others are beyond common understanding or even conflict with traditional Go tactics. This provides human players with new QiGan[1].

**To summarize**, in this paper, we propose to solve distinctive challenges in explaining the Go game. We extend the AND interaction to define the OR interaction, penalize unreliable high-order interactions, prove the universal matching property of AND-OR interactions, and compute the attributions of common coalitions. Professional human players[13] claim that they have learned new QiGan[1] (novel shape patterns beyond current human knowledge) encoded by the value network.

**Extended applications.** Discovering inference patterns encoded by a DNN can be broadly applied to different AI tasks beyond the Go game. In follow-up studies, we have used our method to (1) **explain**

**the Gobang game** and (2) **diagnose incorrect patterns** used by a DNN for object detection. Please see Appendices P.5 and P.6 for details.

## 2 EXPLAINING SHAPE PATTERNS OF STONES

### 2.1 PRELIMINARIES: PROBLEM SETTING AND DEFINITION OF AND INTERACTIONS[3]

Given a game board state $\boldsymbol{x}$ as the input, the value network estimates the probability $p_{\text{white}}(\boldsymbol{x}) \in [0, 1]$ that white stones will win (Silver et al., 2016). Let us use $\boldsymbol{x} = \{x_1, x_2, ..., x_n\}$[5] to denote both the positions and colors of $n$ stones of the current state, which is indexed by $N = \{1, 2, \cdots, n\}$. In this way, the value network's output advantage score of the white stone $v(\boldsymbol{x}) \in \mathbb{R}$ can be defined as

$$v(\boldsymbol{x}) = \log \left( \frac{p_{\text{white}}(\boldsymbol{x})}{1 - p_{\text{white}}(\boldsymbol{x})} \right). \tag{1}$$

**Problem setting.** The basic idea is to construct a logical model $h$ to explain the detailed shape patterns encoded by the value network. Specifically, two key requirements have been imposed to guarantee the faithfulness of the explanation based on the logical model. **(1) Fidelity requirement** stipulates that the logical model $h$ must be capable of accurately predicting all of the DNN's outputs for a sufficiently large input sample set $\Psi$. **(2) Conciseness requirement** demands that the logical model $h$ should be simple enough to encode only a few clear inference patterns used by the DNN. Mathematically, these two requirements are formulated as follows.

$$\forall \boldsymbol{x}' \in \Psi, \quad h(\boldsymbol{x}') = v(\boldsymbol{x}'), \quad \text{subject to} \quad \text{complexity}(h) \leq M, \tag{2}$$

where $M$ is an upper bound on the complexity of the logical model $h$.

**Specifically, Ren et al. (2024) have implemented the above logical model $h$ as a model that encodes AND interaction logic.**

$$\forall \boldsymbol{x}' \in \Psi, \quad h(\boldsymbol{x}') \stackrel{\text{def}}{=} \sum\nolimits_{\emptyset \neq S \in \Omega^{\text{AND}}} \mathbb{1}_{\text{AND}}(S|\boldsymbol{x}') \cdot I_S^{\text{AND}} + b, \tag{3}$$

where the sample $\boldsymbol{x}' \in \Psi$ can be considered as a board state of the Go game in this paper. $b \in \mathbb{R}$ is a scalar bias. The trigger function $\mathbb{1}_{\text{AND}}(S|\boldsymbol{x}') \in \{0, 1\}$ represents an AND interaction among the input variables in $S \subseteq N$. It outputs 1 only when all input variables in $S$ are present in $\boldsymbol{x}'$. Removing/masking any input variable in $S$ will deactivate this interaction, leading to $\mathbb{1}_{\text{AND}}(S|\boldsymbol{x}') = 0$. The scalar weight $I_S^{\text{AND}} \in \mathbb{R}$ is referred to as the numerical effect of the interaction.

**Then, the fidelity requirement is implemented as the *universal matching property* of interactions in Theorem 1.** Ren et al. (2023) have proven that a specific logical model $h$ can precisely mimic all the DNN's outputs $v(\cdot)$, regardless of how variables in the input sample are randomly masked. Specifically, in the context of the Go game, given a board state $\boldsymbol{x}$ with $n$ stones, the logical model's outputs $h(\cdot)$ can always accurately predict the advantage scores $v(\cdot)$ estimated by the value network, no matter how we randomly remove stones from the board. The input sample set $\Psi = \{\boldsymbol{x}_T \mid T \subseteq N\}$ contains all $2^n$ masked board states, where $\boldsymbol{x}_T$ denotes the board state that retains the stones in set $T$ while removing those in $N \setminus T$. Thus, $\Psi$ is sufficiently large to meet the fidelity requirement.

**Theorem 1** (**Universal matching property**)**.** *Let the scalar weights be set to* $\forall S \subseteq N, I_S^{\text{AND}} \stackrel{\text{def}}{=} \sum_{L \subseteq S} (-1)^{|S|-|L|} \cdot v(\boldsymbol{x}_L)$*, and the bias be set to* $b = v(\boldsymbol{x}_\emptyset)$*. Then we have*

$$\forall \boldsymbol{x}' \in \Psi, \quad h(\boldsymbol{x}') = v(\boldsymbol{x}'). \tag{4}$$

**Finally, the conciseness requirement is implemented as the *sparsity property* of interactions.** Ren et al. (2024) take the number of interactions encoded by the logical model $h$ as a measure of its complexity. They have proven that a well-trained DNN typically encodes only $\mathcal{O}(n^\xi/\tau) \ll 2^n$ salient interactions[6], *i.e.,* complexity$(h) \ll 2^n$. The set of salient interactions is defined as $\Omega_{\text{salient}}^{\text{AND}} \stackrel{\text{def}}{=} \{S \subseteq$

---

[5]Although the input to the value network is a tensor (Silver et al., 2016), we use $\boldsymbol{x} = \{x_1, x_2, ..., x_n\}$ to represent the input for simplicity.

[6]Ren et al. (2024) have partially demonstrated the sparsity property under three common conditions. While it is often impractical to strictly verify that a network's inference on a given input sample satisfies these conditions, experimental results in Figure 2 and Figure 5 (b) confirm the sparsity of interactions encoded by the KataGo model. Please see Appendix D for more details. Empirically, $\xi$ is usually in the range of $[0.9, 1.2]$.

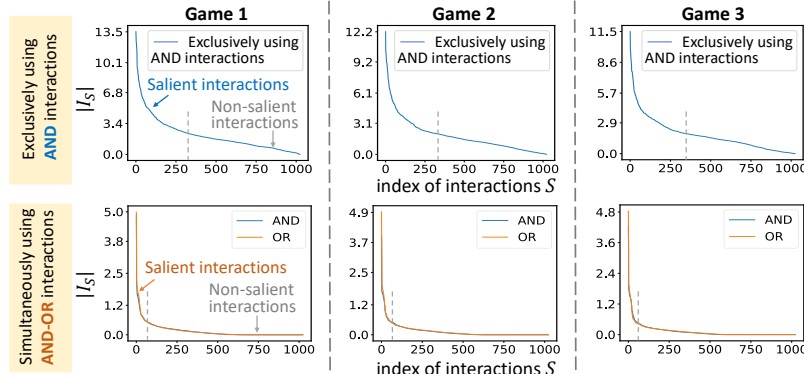

Figure 2: Interaction explanations generated by using both AND interactions and OR interactions are sparser[7] (more concise) than those generated by using exclusively AND interactions. We sort the strength of all interaction effects $|I_S^{\mathrm{AND}}|$ and $|I_S^{\mathrm{OR}}|$ in descending order. Only a few AND-OR interactions have salient effects $|I_S^{\mathrm{AND}}|$ and $|I_S^{\mathrm{OR}}|$.

$N : |I_S^{\mathrm{AND}}| > \tau\}$[7]. All other interactions have negligible numerical effects. Therefore, the logical model $h$ can be constructed solely based on salient interactions while still accurately approximating the outputs of the value network on $\Psi$.

Thus, the aforementioned requirements for fidelity and conciseness enable the interactions to be faithfully regarded as the primitive shape patterns utilized by the DNN[3].

## 2.2 DEFINING OR INTERACTIONS

**Limitations of using AND interactions to explain OR relationships.** Because the Go game involves a high degree of complex logic (Shin et al., 2021), its value network is supposed to encode both AND relationships and OR relationships among stones.

However, we have found that traditional interactions cannot concisely represent OR relationships encoded by the value network. Specifically, Theorem 2 demonstrates that an OR relationship among $m$ variables (stones) can be redundantly explained as exponentially many ($2^m - 1$) AND interactions.

**Theorem 2** (proved in Appendix H). *Consider a function $f$ that encodes a single OR relationship among $m$ variables in $S = \{k_1, k_2, \cdots, k_m\}$. For all randomly masked samples $\boldsymbol{x}' \in \Psi$, the outputs of $f$ can be well matched by an OR trigger function, i.e., $f(\boldsymbol{x}') = \mathbb{1}_{\mathrm{OR}}(S|\boldsymbol{x}') \cdot I_S^{\mathrm{OR}}$, subject to*

$$\mathbb{1}_{\mathrm{OR}}(S|\boldsymbol{x}') = (\mathrm{exist}(x_{k_1}) \vee \mathrm{exist}(x_{k_2}) \vee \cdots \vee \mathrm{exist}(x_{k_m})), \tag{5}$$

*where $\vee$ represents the binary logical operation "OR." The Boolean function $\mathrm{exist}(\cdot) = 0$ when the variable $x_{k_i}$ is masked. If any variable in $S$ exists in the sample $\boldsymbol{x}'$, $\mathbb{1}_{\mathrm{OR}}(S|\boldsymbol{x}') = 1$; otherwise, $\mathbb{1}_{\mathrm{OR}}(S|\boldsymbol{x}') = 0$. Then, the function $f$ can be equivalently explained by $2^m - 1$ different AND interactions as follows.*

$$\forall \boldsymbol{x}' \in \Psi, \quad f(\boldsymbol{x}') = \sum_{\emptyset \neq S' \subseteq S} I_{S'}^{\mathrm{AND}} \cdot \mathbb{1}_{\mathrm{AND}}(S'|\boldsymbol{x}'),$$

$$s.t. \quad \forall S' \subseteq S, S' \neq \emptyset, \quad I_{S'}^{\mathrm{AND}} = I_S^{\mathrm{OR}} \cdot (-1)^{|S'|-1} \neq 0, \tag{6}$$

*where $|S'|$ is the number of variables in $S'$.*

**Extending AND interactions to OR interactions.** Therefore, to solve the redundancy in representing OR relationships, we extend AND interactions to define OR interactions, so that we can simultaneously use both AND interactions and OR interactions to concisely explain the detailed shape patterns encoded by the value network (see Figure 2 for verification). Specifically, the revised logical model $g$ with both AND interactions and OR interactions can be formulated as follows.

$$\forall \boldsymbol{x}' \in \Psi, \quad g(\boldsymbol{x}') \overset{\text{def}}{=} \sum_{\emptyset \neq S \in \Omega^{\mathrm{AND}}} \mathbb{1}_{\mathrm{AND}}(S|\boldsymbol{x}') \cdot I_S^{\mathrm{AND}} + \sum_{\emptyset \neq S \in \Omega^{\mathrm{OR}}} \mathbb{1}_{\mathrm{OR}}(S|\boldsymbol{x}') \cdot I_S^{\mathrm{OR}} + b, \tag{7}$$

---

[7]In contrast with traditional sparsity definitions (Tibshirani, 1996), Ren et al. (2024) define sparsity in terms of non-salient interactions rather than by counting zero values. They typically set a small threshold $\tau = 0.15 \cdot \max_S |I_S^{\mathrm{AND}}|$ to select salient interactions. See Ren et al. (2024) for detailed proofs.

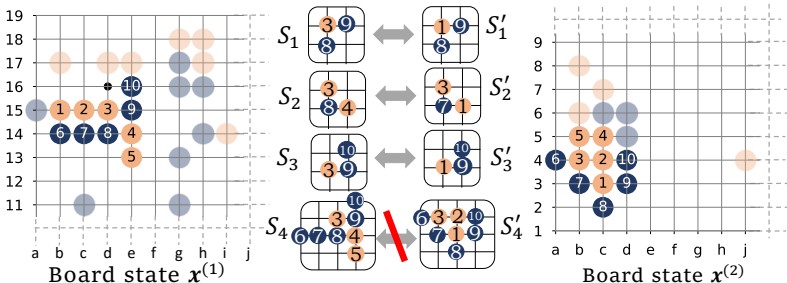

Figure 3: Transferability of interactions. Compared to high-order interactions $S_4$, low-order interactions $S_1, S_2, S_3$ from the board state $\boldsymbol{x}^{(1)}$ are easier to be transferred to another board state $\boldsymbol{x}^{(2)}$.

where scalar weights $I_S^{\text{AND}}$, $I_S^{\text{OR}}$ are derived in Theorem 3 and Equation (9) below.

**We continue to utilize the problem setting in Equation (2) to evaluate the faithfulness of the explanation**. We demonstrate that the newly proposed AND-OR logical model $g$ still satisfies both fidelity and conciseness requirements.

**First, we prove Theorem 3, which shows that under a specific setting of $I_S^{\text{AND}}$, $I_S^{\text{OR}}$, the AND-OR logical model $g$ continues to meet the fidelity requirement.** Specifically, we implement the fidelity requirement by extending the universal matching property to AND-OR interactions. Similar to Theorem 1, the new logical model $g$ accurately mimics all the estimated advantage scores $v(\cdot)$ of the value network, regardless of how stones are randomly removed from the board. The input sample set $\Psi$ is implemented as the $2^n$ masked board states, *i.e.*, $\Psi = \{\boldsymbol{x}_T \mid T \subseteq N\}$.

**Theorem 3** (**Universal matching property of AND-OR interactions**, proved in Appendix G). *Let the scalar weights in the AND-OR logical model $h(\cdot)$ be set to $\forall S \subseteq N, I_S^{\text{AND}} \stackrel{def}{=} \sum_{T \subseteq S}(-1)^{|S|-|T|} \cdot u_T^{\text{AND}}, I_S^{\text{OR}} \stackrel{def}{=} -\sum_{T \subseteq S}(-1)^{|S|-|T|} u_{N\setminus T}^{\text{OR}}$, where $u_T^{\text{AND}} = 0.5 \cdot v(\boldsymbol{x}_T) + \gamma_T$, $u_T^{\text{OR}} = 0.5 \cdot v(\boldsymbol{x}_T) - \gamma_T$. $\{\gamma_T \mid T \subseteq N\}$ is a set of learnable parameters. The scalar bias is set to $b = v(\boldsymbol{x}_\emptyset)$. Then we have*

$$\forall \boldsymbol{x}' \in \Psi, \quad g(\boldsymbol{x}') = v(\boldsymbol{x}'). \tag{8}$$

**Second, we propose an algorithm to train $\{\gamma_T \mid T \subseteq N\}$ such that the learned AND-OR logical model $g$ satisfies the conciseness requirement.** Although Theorem 3 has ensured that the logical model $g$ maintains the universal matching property no matter how we set $\{\gamma_T\}$, we still need to train $\{\gamma_T\}$ to further enhance the sparsity of the extracted interactions. Specifically, we develop the following LASSO-like loss function.

$$\min_{\{\gamma_T\}} \text{Loss}, \quad \text{Loss} = \|\boldsymbol{I}_{\text{AND}}\|_1 + \|\boldsymbol{I}_{\text{OR}}\|_1, \tag{9}$$

where $\|\cdot\|_1$ represents L-1 norm. We vectorize all AND interactions as $\boldsymbol{I}_{\text{AND}} = \text{vec}(\{I_S^{\text{AND}} \mid S \subseteq N\}) \in \mathbb{R}^{2^n}$, and vectorize all OR interactions as $\boldsymbol{I}_{\text{OR}} = \text{vec}(\{I_S^{\text{OR}} \mid S \subseteq N\}) \in \mathbb{R}^{2^n}$. $\boldsymbol{I}_{\text{AND}}$ and $\boldsymbol{I}_{\text{OR}}$ are parameterized by $\{\gamma_T\}$ according to Theorem 3.

**Order of interactions.** The order of an interaction is determined by the number of input variables (in our case, stones) in $S$, denoted as $\text{order}(S) = |S|$. Order reflects the complexity of an interaction.

Theorem 5 in Appendix F demonstrates that even small noise in the value network's outputs $v(\cdot)$ can undermine the stability of higher-order interactions. To address this, we slightly modify the network output from $v(\boldsymbol{x}_T)$ to $v'(\boldsymbol{x}_T) = v(\boldsymbol{x}_T) + \epsilon_T$, where $\epsilon_T \in \mathbb{R}$ is a small scalar bounded by $|\epsilon_T| < \eta$[8], representing inevitable noise in the network output. Based on this adjustment, we redefine the AND-OR interactions using the revised outputs $v'(\boldsymbol{x}_T)$, and the learning of AND-OR interactions can then be reformulated as $\min_{\{\gamma_T\}, \{\epsilon_T\}} \text{Loss}$. Experimental results in Figure 2 have demonstrated the effectiveness of the proposed algorithm. Please see Section 2.4 for details.

## 2.3 HOW TO OBTAIN TRANSFERABLE SHAPE PATTERNS

**High-order interactions are usually less transferable to other boards.** We aim to extract interactions that represent convincing primitive shape patterns, which can generalize (be transferred) to

---

[8]See Appendix O.3 for details on choosing the threshold $\eta$.

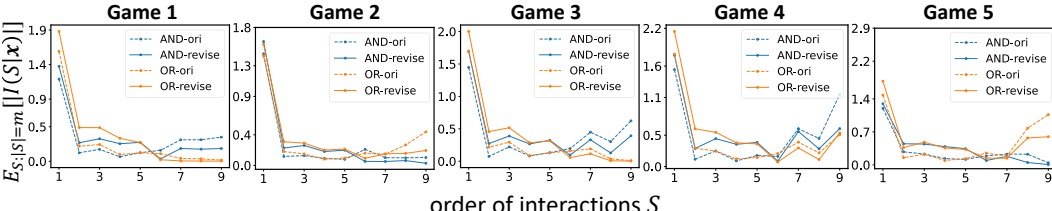

Figure 4: Effectiveness of reducing high-order interactions. We show the average strength of numerical effect for each order of interaction. For each game, the revised method in Equation (10) extracts high-order interactions with weaker effects than the original method in Equation (9).

different game boards. However, Zhou et al. (2024) have pointed out that high-order interactions are less likely to generalize to other inputs. Although their findings are specific to AND interactions, our visualization of AND-OR interactions reveals a similar phenomenon: high-order AND-OR interactions are generally less transferable to other boards than low-order ones.

For example, as Figure 3 shows, 3-order interactions $S_1, S_2, S_3$ extracted from the board $x^{(1)}$ can be transferred to the board $x^{(2)}$. However, the 8-order interaction $S_4$ extracted from the board state $x^{(1)}$ cannot be transferred to the board state $x^{(2)}$. This observation is further validated in Figure 9 in Appendix P.2, where we quantify the proportion of transferable interactions for each order.

**Constraining the order of interactions.** To enhance the transferability of AND-OR interactions, a new loss function that penalizes higher-order interactions is proposed[9].

$$\text{Loss}^{\text{new}} = \|I_{\text{AND}}\|_1 + \|I_{\text{OR}}\|_1 + r \cdot (\|I_{\text{AND}}^{\text{high}}\|_1 + \|I_{\text{OR}}^{\text{high}}\|_1), \tag{10}$$

where $I_{\text{AND}}^{\text{high}}$ and $I_{\text{OR}}^{\text{high}}$ denote the compact vectors corresponding to high-order AND and OR interactions, respectively. Based on the recommendations of professional human players[10], we empirically set $I_{\text{AND}}^{\text{high}}$ as a 386-dimensional vector, corresponding to all interactions from the 6-th to the 10-th order. We set $r = 5.0$ to intensify the penalty on high-order interactions. The below Experiment 1 verifies the effectiveness of our method in penalizing high-order interactions.

**High-order shape patterns $\neq$ global shape patterns.** Human Go players[13] typically use global shape patterns rather than high-order shape patterns in their play. A high-order shape pattern contains most of the $n$ stones. In contrast, a global shape pattern refers to a pattern that occurs across different regions of the board, but it does not necessarily contain most of the $n$ stones.

**Experimental settings.** We conduct our experiments using KataGo v1.13.0 (Wu, 2019), an open-source DNN for Go that has defeated human players. To generate board states, we let KataGo alternately play $\frac{m}{2}$ black and $\frac{m}{2}$ white moves. Since computing interactions among all $m$ stones is NP-hard, we invite *a professional human Go player* to select[11] $n = 10$ representative stones ($n \leq m$), including $\frac{n}{2}$ white and $\frac{n}{2}$ black, and compute interactions involving only these stones. The remaining $(m - n)$ stones are treated as a fixed background and excluded from the interaction extraction.

**Experiment 1: effectiveness of penalizing high-order interactions.** We conduct experiments to check whether the methods in Equation (10) can reduce the complexity (order) of the extracted interactions, compared to the original methods in Equation (9). Specifically, we compute the average strength of AND-OR interactions of different orders, $\mathbb{E}_{S:|S|=m}[|I_S^{\text{AND}}|]$ and $\mathbb{E}_{S:|S|=m}[|I_S^{\text{OR}}|]$, respectively. Figure 4 shows the average strength of interaction effects. For both AND and OR interactions, we observe that the revised method in Equation (10) extracts much weaker high-order (complex) interactions than the original method in Equation (9).

---

[9]Please see Appendix A for the ***pseudo code*** of our complete method.

[10]This is a fully empirical setting. We define interactions of the sixth order and above as high-order, since it is challenging even for expert human players to identify meaningful patterns in interactions beyond the sixth order. Nevertheless, users can select different orders when applying this method to explain other games.

[11]We acknowledge that the choice of stones may slightly affect detected interactions; however, results show our conclusion on sparsity is robust to the specific selection. Involving a professional player also ensures the chosen stones are likely to have meaningful correlations.

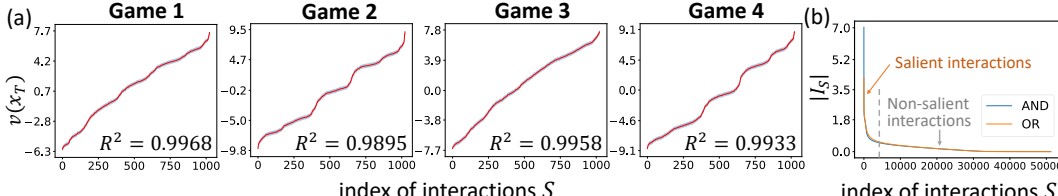

Figure 5: (a) Examination of using the AND-OR interactions to mimic the output of the value network. Outputs of the value network on all $2^n$ masked board states $v(\boldsymbol{x}_T)$ (the red full line) are arranged in ascending order. The height of the blue shade represents the smoothed matching error, computed by averaging the matching error $\Delta v_T = |v(\boldsymbol{x}_T) - g(\boldsymbol{x}_T)|$ of neighboring 50 masked states. We also report the $R^2$ valueKvålseth (1985); Draper & Smith (1998) for the matching $1 - \frac{\sum_T [v(x_T) - g(x_T)]^2}{\sum_T [v(x_T) - E_S v(x_S)]^2}$. (b) Sparsity[7] of the AND-OR interactions. The strengths of all AND and OR interactions from 50 games are ranked in descending order, showing that only a few interactions have salient effects.

**Experiment 2: showing more cases of transferable interactions.** We conduct the second experiment to show that our method successfully ensures a high transferability of interactions across different boards. Please see Figure 8 in Appendix P.1 for details.

### 2.4 FAITHFULNESS OF AND-OR INTERACTION-BASED EXPLANATION

Examining whether the extracted AND-OR interactions faithfully represent the actual shape patterns encoded by the value network is the core issue of this study. Therefore, in this subsection, we demonstrate the faithfulness of our explanation from both theoretical and experimental perspectives.

**Theoretical guarantee.** The faithfulness of newly proposed AND-OR interactions is also theoretically guaranteed by the previous two requirements. (1) Fidelity requirement: all technical extensions in Sections 2.2 and 2.3 follow the paradigm established in Theorem 3, so AND-OR interaction still satisfies the universal matching property, *i.e.*, they can accurately explain the advatange scores on all the $2^n$ masked states in $\{\boldsymbol{x}_T \mid T \subseteq N\}$. (2) Conciseness requirement: the proposed methods described in Equation (9) and Equation (10) ensure the sparsity[7] of AND-OR interactions in the logical model $g$.

In addition, we prove Theorem 4 in Appendix E to show that an OR interaction can be considered as a specific AND interaction. We also prove Theorem 6 in Appendix I to show that AND-OR interactions can serve as the elementary numerical components of the Shapley value (Shapley, 2016).

**Experimental verification.** We conducted experiments to verify the faithfulness of interactions from three perspectives, *i.e.*, conciseness, fidelity, and transferability.

• *Towards the conciseness requirement.* Although Ren et al. (2024) have proved the sparsity property[7] of interactions under certain common conditions[6], it remains challenging to determine whether these conditions are fully satisfied in the context of the Go game. Thus, it is necessary to empirically verify the sparsity[7] of interactions encoded by KataGo v1.13.0 (Wu, 2019).

First, let us examine whether the extension to AND-OR interactions in Equation (9) can successfully boost the sparsity[7] of interactions. The experimental settings are the same as those in Section 2.3. Figure 2 presents the strengths $|I_S^{\mathrm{AND}}|$ and $|I_S^{\mathrm{OR}}|$ of all AND and OR interactions, sorted in descending order. The results show that simultaneously extracting both AND interactions and OR interactions yields a sparser explanation of the value network than relying solely on AND interactions. More than $85\% - 90\%$ of interactions are found to have negligible numerical effects[7].

Then, let us examine the sparsity of the finally extracted AND-OR interactions based on Equation (10). We follow experimental settings in Section 2.3 to generate 50 game states, and visualize the strength of AND interactions and OR interactions of all 50 game states together in a descending order. Figure 5 (b) shows that only a few interactions have salient effects, and more than $90\%$ interactions have small effects, which verifies the sparsity[7] of interactions extracted by the proposed method.

• *Towards the fidelity requirement.* We verify the fidelity requirement by evaluating whether the universal matching property in Theorem 3 is fulfilled by the final proposed method in Equation (10).

To this end, we conduct experiments on a given board state $\boldsymbol{x}$, and we check whether the extracted AND-OR interactions can accurately mimic the network outputs $v(\boldsymbol{x}_T)$ on all $2^n$ randomly masked board states $\{\boldsymbol{x}_T \mid T \subseteq N\}$. To this end, for each masked board states $\boldsymbol{x}_T$, we measure the matching error $\Delta v_T = |v(\boldsymbol{x}_T) - g(\boldsymbol{x}_T)|$ of using AND-OR interactions to match the real output $v(\boldsymbol{x}_T)$, where $g(\boldsymbol{x}_T)$ represents the score predicted by the logical model based on AND-OR interactions. In Figure 5 (a), the solid curve shows the real network outputs on all $2^n$ randomly masked board states in ascending order. The shade area shows the smoothed matching error, which is computed by averaging matching errors of neighboring 50 masked board states. Figure 5 (a) shows that the output of the logical model $g(\boldsymbol{x}_T)$ can well match the real outputs $v(\boldsymbol{x}_T)$ over different randomly masked states, which verifies the fidelity requirement.

• *Towards transferability.* Experiments in Appendix P.1 show the transferability of interactions, *i.e.*, interactions (shape patterns) extracted from one board can also explain $v(\boldsymbol{x}_T)$ in another board.

## 2.5 ILLUSTRATION OF NOVEL SHAPE PATTERNS

With the methods[9] proposed in previous subsections, we typically extract 100–250 salient interactions from each input board state. However, the number of salient interactions is still too large to analyse.

Therefore, we propose a more efficient method for illustrating novel shape patterns. Specifically, we first present all interactions, and then selectively visualize certain common combinations of stones that frequently occur across different interactions. We refer to these combinations as "*common coalitions.*" For example, Figure 1 shows that given a board state $\boldsymbol{x}$ with $n$ stones, $N = \{1, 2, ..., n\}$, we can extract some salient interactions from the board state $\boldsymbol{x}$, such as $S_1 = \{1, 2, 3, 6\}, S_2 = \{1, 2, 3, 6, 7\}$, $S_3 = \{1, 2, 3, 8\}$, *etc.* The coalition $T = S_1 \cap S_2 \cap S_3 = \{1, 2, 3\}$ occurs in each interaction, so we can consider this common coalition $T$ as a *shape pattern* encoded by the value network.

To better understand the influence of common coalitions on the Go game, we further compute the attribution $\varphi(T)$ of each coalition $T$ to the advantage score $v(\boldsymbol{x})$ estimated by the value network. A positive or negative value of $\varphi(T)$ indicates that the shape pattern formed by coalition $T$ tends to increase or decrease the advantage score of white stones. Although there are many attribution methods (Lundberg & Lee, 2017; Selvaraju et al., 2017; Zhou et al., 2016; Zintgraf et al., 2017), (Xinhao Zheng, 2025) have extended the attribution of individual variables to the attribution of a coalition $T$. Thus, we apply this method to generate a self-consistent[12] attribution for the coalition $T$. Specifically, the attribution score $\varphi(T)$ of the coalition $T$ is formulated as the weighted sum of the effects of AND-OR interactions[9].

$$\varphi(T) = \sum\nolimits_{S \supseteq T} (|T|/|S|) \cdot (I_S^{\mathrm{AND}} + I_S^{\mathrm{OR}}). \tag{11}$$

In other words, given a set of AND interactions and OR interactions, the attribution of the coalition $\varphi(T)$ can be computed by allocating a ratio $\frac{|T|}{|S|}$ of the effect of each interaction $S \subseteq T$ containing the coalition $T$. *In addition, Appendix M shows a list of desirable theorems and properties of the attribution metric $\varphi(T)$, which theoretically guarantee the faithfulness of the attribution metric $\varphi(T)$.*

**Experiments.** Given a board state, we extract two sets of salient AND-OR interactions $\Omega_{\mathrm{salient}}^{\mathrm{AND}} = \{S \subseteq N : |I_S^{\mathrm{AND}}| > \tau\}$ and $\Omega_{\mathrm{salient}}^{\mathrm{OR}} = \{S \subseteq N : |I_S^{\mathrm{OR}}| > \tau\}$, where $\tau = 0.15 \cdot \max_S\{|I_S^{\mathrm{AND}}|, |I_S^{\mathrm{OR}}|\}$. We manually annotate 50 coalitions shared by these interactions based on the guidance from professional human Go players[13]. Figure 6 visualizes sixteen coalitions selected from four game states.

## 2.6 HUMAN INTERPRETATION OF THE SHAPE PATTERNS[9]

In order to explain shape patterns (common coalitions) encoded by the value network, we collaborate with a professional human Go player[13]. Based on Figure 6, they find that some shape patterns fit QiGan[1] of human players, and other shape patterns conflict with human understandings (QiGan[1]). They acknowledge gaining substantial new insights into the game from the extracted shape patterns.

**Shape patterns that fit traditional human understandings (traditional QiGan[1]). For the Game 1** in Figure 6 (1.a - 1.d), $\varphi(\{1, 2, 3, 8\}) < \varphi(\{2, 3, 8\})$ and $\varphi(\{1, 2, 3, 7\}) < \varphi(\{2, 3, 7\})$. It means that

---

[12]Please see Appendix N for details about the self-consistency property of attributions.

[13]During the review phase, the Go players are anonymous, because one of them also serves as an author. Please see Appendix L for the composition of the Go players.

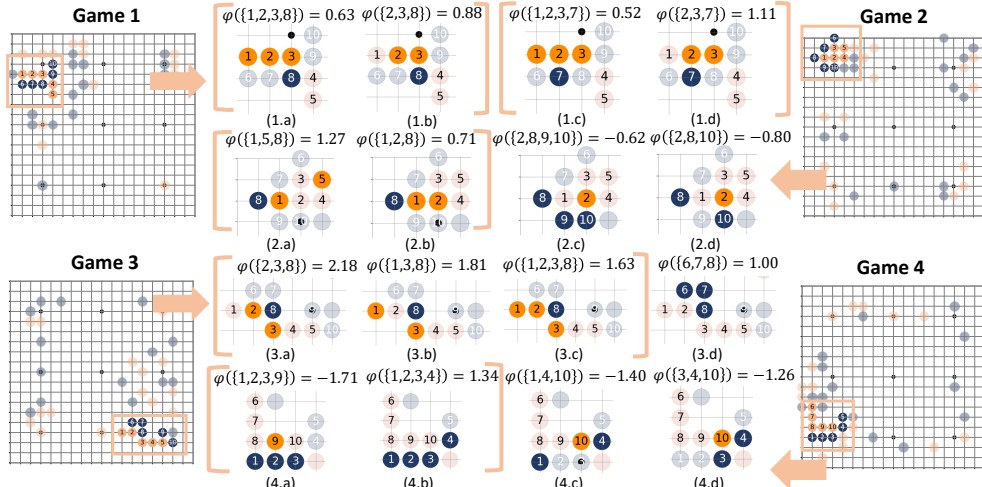

Figure 6: Estimated attributions of different coalitions (shape patterns).

when the white stone $x_1$ participates in the combination of white stones $x_2, x_3$, the advantage of white stones becomes lower, *i.e.*, the stone $x_1$ is a low-value move. Go players also consider that the effect of the combination of white stones $x_1, x_2, x_3$ is low. **For the Game 2** in Figure 6 (2.a, 2.b), $\varphi(\{1,2,8\}) < \varphi(\{1,5,8\})$ means that the value network considers that the white stone $x_2$ has lower value than $x_5$. Go players consider that in this game state, the white stone $x_5$ protects the white stones $x_1, x_2, x_3, x_4$, and the white stones $x_1, x_3$ attack the black stones $x_6, x_7$, but the white stone $x_2$ has much less value than other stones. **For the Game 3** in Figure 6 (3.a - 3.c), $\varphi(S_1 = \{1,3,8\}) > \varphi(S_3 = \{1,2,3,8\})$ and $\varphi(S_2 = \{2,3,8\}) > \varphi(S_3 = \{1,2,3,8\})$, subject to $S_3 = S_1 \cup S_2$. It means that given the context $S_1$, the stone $x_2$ wastes a move, and given the context $S_2$, the stone $x_1$ wastes a move. Go players also consider that the combination of stones $x_1, x_2$ is of low value.

**Shape patterns that conflict with human understandings (QiGan[1]). For the Game 3** in Figure 6 (3.d), Go players find some cases that conflict with their professional understanding. For example, $\varphi(\{6,7,8\}) = 1.00$, it means that the value network considers that the coalition $\{6,7,8\}$ is advantageous for white stones. However, there are 3 black stones and no white stones in $\{6,7,8\}$, Go players consider that this coalition is advantageous for black stones. **For the Game 4** in Figure 6 (4.b), $\varphi(\{1,2,3,4\}) = 1.34$. It means that the value network considers that the coalition $\{1,2,3,4\}$ is advantageous for white stones. However, there are four black stones in this coalition, and Go players think that the coalition is advantageous for black stones.

*Please see Appendix P.3 and Figure 10 for an analysis on more boards with lots of new understanding.*

## 3 CONCLUSION AND DISCUSSIONS

In this paper, we extract sparse interactions among stones that the value network memorizes for the game of Go. We define the OR interactions and propose a method to extract sparse AND-OR interactions to represent the shape patterns. The universal matching property of AND-OR interactions is proven to guarantee the faithfulness of the explanation.

Then, we examine how the automatically extracted shape patterns fit or conflict with conventional human understanding of the Go, so as to illustrate distinct QiGan[1] encoded by the value network to play the Go game. We collaborate with professional human Go players to analyze the QiGan[1] that is automatically extracted from the value network. They report that they have gained some new insights.

**Board applicability: using AND-OR interactions to explain DNNs for other AI tasks.** Our explanation method[9] is a general approach to disentangling inference patterns encoded by DNNs. To further showcase its practical utility beyond the Go game, we illustrate how interactions can be used to (1) **explain the Gobang game** and (2) **diagnose incorrect patterns** used by a DNN for object detection, as discussed in Appendices P.5 and P.6.

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

## A  PSEUDOCODE OF OUR METHODS

To clarify how we explain the shape patterns encoded by the value network, we show the pseudocode of our complete method (see Appendices J and K for detailed optimization techniques).

---

**Algorithm 1** Disentangling the shape patterns encoded by the value network $v(\cdot)$

---

**Input:** a board state $x$, a value network $v(\cdot)$.
**Output:** all AND interactions $I_S^{\mathrm{AND}}$ and all OR interactions $I_S^{\mathrm{OR}}$, *w.r.t.* $\forall S \subseteq N$, and the attributions $\varphi(T)$ of each coalition $T$.
**Step 1:** initializing parameters.

$\quad \forall k \in \{-\frac{n}{2}, -\frac{n}{2}+1, ..., \frac{n}{2}\}$, set $a_k = \mathbb{E}_{\boldsymbol{x}}\mathbb{E}_{T \subseteq N : \Delta n(T)=k} \log(\frac{p_{\mathrm{white}}(\boldsymbol{x}_T)}{1-p_{\mathrm{white}}(\boldsymbol{x}_T)})$.

$\quad \forall T \subseteq N$, set $\gamma_T = \epsilon_T = 0$.

**Step 2:** iteratively updating parameters $\boldsymbol{a} = \{a_{-\frac{n}{2}}, a_{-\frac{n}{2}+1}, ..., a_{\frac{n}{2}}\}, \{\gamma_T \mid T \subseteq N\}, \{\epsilon_T \mid T \subseteq N\}$.

$\quad \min\limits_{\boldsymbol{a},\{p_T\},\{q_T\}} \|\boldsymbol{I}_{\mathrm{AND}}\|_1 + \|\boldsymbol{I}_{\mathrm{OR}}\|_1 + r \cdot (\|\boldsymbol{I}_{\mathrm{AND}}^{\mathrm{high}}\|_1 + \|\boldsymbol{I}_{\mathrm{OR}}^{\mathrm{high}}\|_1)$ based on Equation (10).

**Step 3:** identifying a set of salient interactions $\Omega_{\mathrm{salient}}^{\mathrm{AND}}, \Omega_{\mathrm{salient}}^{\mathrm{OR}}$.

$\quad \Omega_{\mathrm{salient}}^{\mathrm{AND}} = \{S \subseteq N : |I_S^{\mathrm{AND}}| > \tau\}$,

$\quad \Omega_{\mathrm{salient}}^{\mathrm{OR}} = \{S \subseteq N : |I_S^{\mathrm{OR}}| > \tau\}$, where $\tau = 0.15 \cdot \max_S\{|I_S^{\mathrm{AND}}|, |I_S^{\mathrm{OR}}|\}$.

**Step 4:** annotating common coalitions.

$\quad$ Annotate 50 common coalitions $T$ based on salient interactions $\Omega_{\mathrm{salient}}^{\mathrm{AND}}$ and $\Omega_{\mathrm{salient}}^{\mathrm{OR}}$.

**Step 5:** computing coalition attributions.

$\quad$ For each annotated coalition $T$, $\varphi(T) = \sum_{S \supseteq T} \frac{|T|}{|S|}(I_S^{\mathrm{AND}} + I_S^{\mathrm{OR}})$.

---

## B  RELATED WORK

Many methods have been proposed to visualize the feature/patterns encoded by the DNN (Simonyan et al., 2014; Dosovitskiy & Brox, 2016; Yosinski et al., 2015; Zeiler & Fergus, 2014), to estimate the attribution/attention on each input variable (Lundberg & Lee, 2017; Selvaraju et al., 2017; Zhou et al., 2016; Zintgraf et al., 2017). However, the complexity of the Go game presents a new requirement, *i.e.*, accurately extracting exact shape patterns encoded by the DNN. Clarifying exact inference patterns is of significant value in knowledge discovery from DNNs towards various tasks. For example, our method can also be used to discover novel patterns for medical diagnosis.

However, explaining exact shape patterns encoded by a DNN proposes higher requirements for the faithfulness/fidelity of the explanation. The faithfulness is even supposed to be theoretically guaranteed and experimentally verified, beyond a specious understanding. To this end, (1) Ren et al. (2023) and Li & Zhang (2023) have found that a well-trained DNN usually encodes a small number of interactions, and the output score of the DNN on a certain input sample can always be well mimicked by numerical effects of a few salient interactions, no matter how the input sample is randomly masked[6]. (2) Li & Zhang (2023) have further found the considerable transferability of interactions over different samples and over different DNNs. (3) interactions (the AND interaction) can explain the elementary mechanism of previous explanation metrics, *e.g.*, the Shapley value (Shapley, 2016), the Shapley interaction index (Grabisch & Roubens, 1999), and the Shapley Taylor interaction index (Sundararajan et al., 2020).

Despite of above findings, explaining the DNN for the Go game still proposes new challenges. To this end, we extend the AND interaction to the OR interaction, alleviate high-order interactions caused by the value shift of the advantage score, and compute attributions of common coalitions shared by different interactions, thereby obtaining a concise and accurate explanation for shape patterns in the value network.

**Connection between QiGan and the concept of abstraction in cognitive science.** Recent workHo et al. (2019) in cognitive science regards abstraction as the selective compression of state and temporal structure that supports efficient generalization and planning. QiGan operates in the same spirit: by distilling a set of interactions into a set of reusable shape patterns (knowledge), it constructs a tractable

"world model" for the value network to perform a fast situation assessment of the current state of the board while maintaining strategic fidelity.

**Comparison with mechanistic Interpretability.** Mechanistic interpretabilityOlah (2022) treats a neural network as a collection of distinct components rather than a single black box. It studies the roles of individual neurons, attention heads, or subnetworks, and how these parts work together to produce the model's overall behaviorNanda et al.; 2023); Zhou et al. (2018); Meng et al. (2022); Liu et al. (2023); Lieberum et al. (2023); Wang et al. (2023); Elhage et al. (2021); Gao et al. (2025); Geva et al. (2021). In contrast, interaction-based methods treat the *entire* deep neural network as a single black-box model, seeking to explain the *entire* model's internal inference logic through a small number of salient interactions in a post-hoc manner. Unlike mechanistic interpretability, interaction-based explanations do not attempt to interpret the representations of specific neurons within the network. And there is no correspondence between interactions and specific neural network structures.

## C  PROPERTIES FOR THE HARSANYI DIVIDEND

In this paper, we follow Ren et al. (2023) to use the Harsanyi dividend (or Harsanyi interaction) to measure the numerical effect $I(S)$ of the AND interaction containing input variables in $S$. Ren et al. (2023) have proved that the Harsanyi dividend satisfied the following properties, including the *efficiency, linearity, dummy, symmetry, anonymity, recursive, interaction distribution properties*.

(1) Efficiency property: The inference score of a well-trained model $v(\boldsymbol{x})$ can be disentangled into the numerical effects of different interactions $I(S), S \subseteq N$, *i.e.*, $v(\boldsymbol{x}) = \sum_{S \subseteq N} I(S)$.

(2) Linearity property: If the inference score of the model $w$ is computed as the sum of the inference score of the model $u$ and the inference score of the model $v$, *i.e.*, $\forall S \subseteq N, w(\boldsymbol{x}_S) = u(\boldsymbol{x}_S) + v(\boldsymbol{x}_S)$, then the interactive effect of $S$ on the model $w$ can be computed as the sum of the interaction effect of $S$ on the model $u$ and that on the model $v$, *i.e.*, $\forall S \subseteq N, I_w(S) = I_u(S) + I_v(S)$.

(3) Dummy property: If the input variable $i$ is a dummy variable, *i.e.*, $\forall S \subseteq N \setminus \{i\}, v(\boldsymbol{x}_{S \cup \{i\}}) = v(\boldsymbol{x}_S) + v(\boldsymbol{x}_{\{i\}})$, then the input variable $i$ has no interaction with other input variables, *i.e.*, $\forall \emptyset \neq S \subseteq N \setminus \{i\}, I(S \cup \{i\}) = 0$.

(4) Symmetry property: If the input variable $i \in N$ and the input variable $j \in N$ cooperate with other input variables in $S \subseteq N \setminus \{i, j\}$ in the same way, *i.e.*, $\forall S \subseteq N \setminus \{i, j\}, v(\boldsymbol{x}_{S \cup \{i\}}) = v(\boldsymbol{x}_{S \cup \{j\}})$, then the input variable $i$ and the input variable $j$ have the same interactive effect, *i.e.*, $\forall S \subseteq N \setminus \{i, j\}, I(S \cup \{i\}) = I(S \cup \{j\})$.

(5) Anonymity property: If a random permutation $\pi$ is added to $N$, then $\forall S \subseteq N, I_v(S) = I_{\pi v}(\pi S)$ is always guaranteed, where the new set of input variables $\pi S$ is defined as $\pi S = \{\pi(i), i \in S\}$, the new model $\pi v$ is defined as $(\pi v)(\boldsymbol{x}_{\pi S}) = v(\boldsymbol{x}_S)$. This suggests that permutation does not change the interactive effects.

(6) Recursive property: The interactive effects can be calculated in a recursive manner. For $\forall i \in N, S \subseteq N \setminus \{i\}$, the interactive effect of $S \cup \{i\}$ can be computed as the difference between the interactive effect of $S$ with the presence of the variable $i$ and the interactive effect of $S$ with the absence of the variable $i$. *I.e.*, $\forall i \in N, S \subseteq N \setminus \{i\}, I(S \cup \{i\}) = I(S|i \text{ is consistently present}) - I(S)$, where $I(S|i \text{ is consistently present}) = \sum_{L \subseteq S} (-1)^{|S|-|L|} v(\boldsymbol{x}_{L \cup \{i\}})$.

(7) Interaction distribution property: This property describes how an interaction function (Sundararajan et al., 2020) distributes interactions. An interaction function $v_T$ parameterized by a context $T$ is defined as follows. $\forall S \subseteq N$, if $T \subseteq S$, then $v_T(\boldsymbol{x}_S) = c$; if not, $v_T(\boldsymbol{x}_S) = 0$. Then, the interactive effects for an interaction function $v_T$ can be computed as $I(T) = c$, and $\forall S \neq T, I(S) = 0$.

## D  COMMON CONDITIONS FOR THE SPARSITY OF INTERACTIONS ENCODED BY A DNN

Ren et al. (2024) have proved that under some common conditions, a DNN usually only encodes a small number of interactions for inference, *i.e.*, (1) The high-order derivatives of the model output

with respect to the input variables are all zero. (2) The AI model can be used on masked/occluded samples, and when the input sample is less masked/occluded, the AI model will yield a higher confidence score on this sample. (3) The confidence score of the AI model on masked/occluded samples does not significantly degrade.

# E  PROVING THAT THE OR INTERACTIONS CAN BE CONSIDERED AS A SPECIFIC AND INTERACTION

**Theorem 4.** *The OR interaction effect between a set $S$ of stones based on $v(\boldsymbol{x}_T)$, can be computed as a specific AND interaction effect based on the dual function $v'(\boldsymbol{x}_T)$. For $v'(\boldsymbol{x}_T)$, original present stones in $T$ (based on $v(\boldsymbol{x}_T)$) are considered as being removed, and original removed stones in $N \setminus T$ (based on $v(\boldsymbol{x}_T)$) are considered as being present.*

• **proof:** The effect $I_{\text{OR}}(S)$ of an OR interaction $S$ is defined as follows.

$$\forall S \subseteq N, S \neq \emptyset, I_{\text{OR}}(S) = -\sum_{T \subseteq S} (-1)^{|S|-|T|} v(\boldsymbol{x}_{N \setminus T})$$

Here, $\boldsymbol{x}_{N \setminus T}$ denotes the masked board state where stones in the set $N \setminus T$ are placed on the board, and stones in the set $T$ are removed. We reconsider the definition of the masked board state $\boldsymbol{x}$ as the definition of $\boldsymbol{x}'$. In comparison, $\boldsymbol{x}'_T$ denotes the masked board state where stones in the set $T$ are removed (based on the definition of $\boldsymbol{x}_T$, stones in the set $T$ are placed on the board), and stones in the set $N \setminus T$ are placed on the board (based on $\boldsymbol{x}_T$, stones in the set $N \setminus T$ are removed).

In this way, $\boldsymbol{x}_{N \setminus T}$ denotes the same board state as $\boldsymbol{x}'_T$. The effect $I_{\text{OR}}(S|\boldsymbol{x})$ of an OR interaction based on the definition of $\boldsymbol{x}$ can be reformulated as the effect $I'_{\text{AND}}(S|\boldsymbol{x}')$ of an AND interaction based on the definition of $\boldsymbol{x}'$ as follows.

$$I_{\text{OR}}(S|\boldsymbol{x}) = -\sum_{T \subseteq S} (-1)^{|S|-|T|} v(\boldsymbol{x}_{N \setminus T}), \quad S \neq \emptyset$$

$$= -\sum_{T \subseteq S} (-1)^{|S|-|T|} v(\boldsymbol{x}'_T), \quad S \neq \emptyset$$

$$= -I'_{\text{AND}}(S|\boldsymbol{x}'), \quad S \neq \emptyset$$

Therefore, we consider the OR interaction as a specific AND interaction.

For example, we represent the effect of an AND interaction $S = \{1, 2, 3\}$ as $I_S^{\text{AND}} \cdot [\text{exist}(x_1) \wedge \text{exist}(x_2) \wedge \text{exist}(x_3)]$. In comparison, the effect of an OR interaction $S = \{1, 2, 3\}$ is represented as $I_S^{\text{OR}} \cdot [\text{exist}(x_1) \vee \text{exist}(x_2) \vee \text{exist}(x_3)] = I_S^{\text{OR}} \cdot \{\neg[(\neg\text{exist}(x_1)) \wedge (\neg\text{exist}(x_2)) \wedge (\neg\text{exist}(x_3))]\}$, where $\vee$ represents the binary logical operation "OR." The Boolean function $\text{exist}(\cdot) = 0$ when the stone is removed.

# F  PROVING THAT UNAVOIDABLE NOISES IN NETWORK OUTPUT WILL BE ENLARGED IN INTERACTIONS

Actually, the real data inevitably contains some small noises/variations, such as texture variations and the shape deformation in object classification. Therefore, the network output $v(\boldsymbol{x}_T)$ also contains some unavoidable noise.

**Theorem 5.** *Let $Var[v(\boldsymbol{x}_T)]$ denote the variance of the network output $v(\boldsymbol{x}_T)$, and let $Var[v(\boldsymbol{x}_{N \setminus T})]$ denote the variance of the network output $v(\boldsymbol{x}_{N \setminus T})$. If we assume that different masked input samples are independent of each other and have no correlation, then we can derive that the variance of the AND interaction and the variance of the OR interaction will be enlarged.*

$$Var[I_S^{\text{AND}}] = Var[\sum_{T \subseteq S} (-1)^{|S|-|T|} v(\boldsymbol{x}_T)] = \sum_{T \subseteq S} Var[v(\boldsymbol{x}_T)]$$

$$Var[I_S^{\text{OR}}] = Var[-\sum_{T \subseteq S} (-1)^{|S|-|T|} v(\boldsymbol{x}_{N \setminus T})] = \sum_{T \subseteq S} Var[v(\boldsymbol{x}_{N \setminus T})]$$

As Theorem 5 shows, the variance of the masked input sample will enlarge the variance of the AND interactions and the variance of the OR interactions. Therefore, we prove that unavoidable noises in network output will be enlarged in interactions.

## G  PROVING THAT THE NETWORK OUTPUT CAN BE REPRESENTED AS AND-OR INTERACTIONS

**Theorem 3.** *Let the scalar weights in the AND-OR logical model $h(\cdot)$ be set to $\forall S \subseteq N, I_S^{\mathrm{AND}} \overset{def}{=} \sum_{T \subseteq S}(-1)^{|S|-|T|} \cdot u_T^{\mathrm{AND}}, I_S^{\mathrm{OR}} \overset{def}{=} -\sum_{T \subseteq S}(-1)^{|S|-|T|}u_{N \setminus T}^{\mathrm{OR}}$, where $u_T^{\mathrm{AND}} = 0.5 \cdot v(\boldsymbol{x}_T) + \gamma_T$, $u_T^{\mathrm{OR}} = 0.5 \cdot v(\boldsymbol{x}_T) - \gamma_T$. $\{\gamma_T \mid T \subseteq N\}$ is a set of learnable parameters. The scalar bias is set to $b = v(\boldsymbol{x}_\emptyset)$. Then we have*

$$\forall \boldsymbol{x}' \in \Psi, \quad g(\boldsymbol{x}') = v(\boldsymbol{x}'). \tag{12}$$

• **proof:** We derive that for all $2^n$ randomly masked sample $\boldsymbol{x}_T$, the output score $v(\boldsymbol{x}_T)$ of the DNN on $\boldsymbol{x}_T$ can be approximated by the sum of effects of AND-OR interactions, *i.e.*, $v(\boldsymbol{x}_T) = v(\boldsymbol{x}_\emptyset) + \sum_{S \subseteq T, S \neq \emptyset} I_S^{\mathrm{AND}} + \sum_{S \cap T \neq \emptyset, S \neq \emptyset} I_S^{\mathrm{OR}}$

$$
\begin{aligned}
\sum_{S \subseteq T} I_S^{\mathrm{AND}} &= \sum_{S \subseteq T} \sum_{L \subseteq S} (-1)^{|S|-|L|} u_L^{\mathrm{AND}} \\
&= \sum_{L \subseteq T} \sum_{S: L \subseteq S \subseteq T} (-1)^{|S|-|L|} u_L^{\mathrm{AND}} \\
&= \underbrace{u_T^{\mathrm{AND}}}_{L=T} + \sum_{L \subseteq T, L \neq T} u_L^{\mathrm{AND}} \cdot \underbrace{\sum_{m=0}^{|T|-|L|}(-1)^m}_{=0} \\
&= u_T^{\mathrm{AND}} \\
\sum_{S \cap T \neq \emptyset, S \neq \emptyset} I_S^{\mathrm{OR}} &= -\sum_{S \cap T \neq \emptyset, S \neq \emptyset} \sum_{L \subseteq S}(-1)^{|S|-|L|} u_{N \setminus L}^{\mathrm{OR}} \\
&= -\sum_{L \subseteq N} \sum_{S: S \cap T \neq \emptyset, S \supseteq L}(-1)^{|S|-|L|} u_{N \setminus L}^{\mathrm{OR}} \\
&= -\underbrace{u_\emptyset^{\mathrm{OR}}}_{L=N} - \underbrace{u_T^{\mathrm{OR}}}_{L=N \setminus T} \cdot \underbrace{\sum_{|S_2|=1}^{|T|} C_{|T|}^{|S_2|}(-1)^{|S_2|}}_{=-1} \\
&\quad - \sum_{L \cap T \neq \emptyset, L \neq N} u_{N \setminus L}^{\mathrm{OR}} \cdot \underbrace{\sum_{S_1 \subseteq N \setminus T \setminus L} \sum_{|S_2|=|T \cap L|}^{|T|} C_{|T|-|T \cap L|}^{|S_2|-|T \cap L|}(-1)^{|S_1|+|S_2|}}_{=0} \\
&\quad - \sum_{L \cap T = \emptyset, L \neq N \setminus T} u_{N \setminus L}^{\mathrm{OR}} \cdot \underbrace{\sum_{S_2 \subsetneq T} \sum_{S_1 \subseteq N \setminus T \setminus L}(-1)^{|S_1|+|S_2|}}_{=0} \\
&= u_T^{\mathrm{OR}} - u_\emptyset^{\mathrm{OR}}
\end{aligned}
$$

Therefore, $u_T^{\mathrm{OR}} = \sum_{S \cap T \neq \emptyset, S \neq \emptyset} I_S^{\mathrm{OR}} + u_\emptyset^{\mathrm{OR}}$. In this way, we can derive that the output score $v(\boldsymbol{x}_T)$ of the DNN on $\boldsymbol{x}_T$ can be approximated by the sum of effects of AND-OR interactions.

$$
\begin{aligned}
v(\boldsymbol{x}_T) &= u_T^{\mathrm{AND}} + u_T^{\mathrm{OR}} \\
&= \sum_{S \subseteq T} I_S^{\mathrm{AND}} + \sum_{S \cap T \neq \emptyset, S \neq \emptyset} I_S^{\mathrm{OR}} + u_\emptyset^{\mathrm{OR}} \\
&= \sum_{S \subseteq T, S \neq \emptyset} I_S^{\mathrm{AND}} + u_\emptyset^{\mathrm{AND}} + \sum_{S \cap T \neq \emptyset, S \neq \emptyset} I_S^{\mathrm{OR}} + u_\emptyset^{\mathrm{OR}} \\
&= \sum_{S \subseteq T, S \neq \emptyset} I_S^{\mathrm{AND}} + \sum_{S \cap T \neq \emptyset, S \neq \emptyset} I_S^{\mathrm{OR}} + v(\boldsymbol{x}_\emptyset)
\end{aligned}
$$

# H  PROVING THAT EXCLUSIVELY USING AND INTERACTIONS TO EXPLAIN THE OR RELATIONSHIP WILL SIGNIFICANTLY COMPLICATE THE EXPLANATION

**Theorem 2.** *Consider a function $f$ that encodes a single OR relationship among $m$ variables in $S = \{k_1, k_2, \cdots, k_m\}$. For all randomly masked samples $\boldsymbol{x}' \in \Psi$, the outputs of $f$ can be well matched by an OR trigger function, i.e., $f(\boldsymbol{x}') = \mathbb{1}_{\mathrm{OR}}(S|\boldsymbol{x}') \cdot I_S^{\mathrm{OR}}$, where*

$$\mathbb{1}_{\mathrm{OR}}(S|\boldsymbol{x}') = (\mathrm{exist}(x_{k_1}) \vee \mathrm{exist}(x_{k_2}) \vee \cdots \vee \mathrm{exist}(x_{k_m})). \tag{13}$$

*$\vee$ represents the binary logical operation "OR". The Boolean function $\mathrm{exist}(\cdot) = 0$ when the variable $x_{k_i}$ is masked. If any variable in $S$ exists in the sample $\boldsymbol{x}'$, $\mathbb{1}_{\mathrm{OR}}(S|\boldsymbol{x}') = 1$; otherwise, $\mathbb{1}_{\mathrm{OR}}(S|\boldsymbol{x}') = 0$. Then, the function $f$ can be equivalently explained by $2^m - 1$ different AND interactions as follows.*

$$\forall \boldsymbol{x}' \in \Psi, \quad f(\boldsymbol{x}') = \sum\nolimits_{\emptyset \neq S' \subseteq S} I_{S'}^{\mathrm{AND}} \cdot \mathbb{1}_{\mathrm{AND}}(S'|\boldsymbol{x}'),$$

$$s.t. \quad \forall S' \subseteq S, S' \neq \emptyset, \quad I_{S'}^{\mathrm{AND}} = I_S^{\mathrm{OR}} \cdot (-1)^{|S'|-1} \neq 0, \tag{14}$$

*where $|S'|$ is the number of variables in $S'$.*

• **proof:** First, $f(\boldsymbol{x}')$ can be equivalently written as

$$f(\boldsymbol{x}') = \mathbb{1}_{\mathrm{OR}}(S|\boldsymbol{x}') \cdot I_S^{\mathrm{OR}} = I_S^{\mathrm{OR}} \cdot \left[ 1 - \prod_{i \in S}(1 - \mathrm{exist}(x_i)) \right].$$

We attempt to represent $f$ using AND interactions. According to Theorem 1, the scalar weights associated with AND interactions are given by

$$\forall \emptyset \neq S' \subseteq S, \quad I_{S'}^{\mathrm{AND}} = \sum_{L \subseteq S'} (-1)^{|S'|-|L|} f(\boldsymbol{x}_L),$$

where $\boldsymbol{x}_L$ is the sample with variables in $L$ present (*i.e.*, $x_i = 1$ for $i \in L$) and all others absent.

Substituting the form of $f$,

$$I_{S'}^{\mathrm{AND}} = I_S^{\mathrm{OR}} \cdot \sum_{L \subseteq S'} (-1)^{|S'|-|L|} \left[ 1 - \prod_{i \in S}(1 - \mathrm{exist}(x_i)) \right]$$

$$= I_S^{\mathrm{OR}} \cdot \sum_{L \subseteq S'} (-1)^{|S'|-|L|} \left[ 1 - \prod_{i \in L}(1 - \underbrace{\mathrm{exist}(x_i)}_{1}) \cdot \prod_{i \in S \setminus L}(1 - \underbrace{\mathrm{exist}(x_i)}_{0}) \right]$$

$$= 0 + I_S^{\mathrm{OR}} \cdot \sum_{\emptyset \neq L \subseteq S'} (-1)^{|S'|-|L|}$$

$$= I_S^{\mathrm{OR}} \cdot (-1)^{|S'|-1} \neq 0.$$

Therefore, in order to faithfully represent the OR relationship in the model, it is necessary to extend the explanatory framework from AND interactions to explicitly include OR interactions. Otherwise, the interaction-based explanation would be unnecessarily complex and redundant.

# I  PROVING THAT THE SHAPLEY VALUE CAN BE EXPLAINED AS AND-OR INTERACTIONS

**Theorem 6.** *The Shapley value $\phi(i)$ of each input variable $i \in N$ can be explained as a re-allocation of AND-OR interactions, i.e., $\phi(i) = \sum_{S \subseteq N, S \ni i} \frac{1}{|S|} I_S^{\mathrm{AND}} + \sum_{S \subseteq N, S \ni i} \frac{1}{|S|} I_S^{\mathrm{OR}}$.*

• **proof:** The Shapley value of the input varibale $i$ is defined as follows. $\phi(i) = \mathbb{E}_{S \subseteq N \setminus \{i\}}[v(\boldsymbol{x}_{S \cup \{i\}}) - v(\boldsymbol{x}_S)]$. For simplicity, we use $v(S)$ to represent the advantage score $v(\boldsymbol{x}_S)$

on the masked board state $\boldsymbol{x}_S$. In the proof of this theorem, we denote $I_S^{\text{AND}}$ as $I_{\text{AND}}(S)$, and denote $I_S^{\text{OR}}$ as $I_{\text{OR}}(S)$.

According to Theorem 3, $\forall S \subseteq N, v(S) = v(\emptyset) + \sum_{T \subseteq S, T \neq \emptyset} I_{\text{AND}}(T) + \sum_{T \cap S \neq \emptyset} I_{\text{OR}}(T)$. Thus,

$$v(S \cup \{i\}) - v(S)$$

$$= \left[ v(\emptyset) + \sum_{T \subseteq (S \cup \{i\}), T \neq \emptyset} I_{\text{AND}}(T) + \sum_{T \cap (S \cup \{i\}) \neq \emptyset} I_{\text{OR}}(T) \right]$$

$$- \left[ v(\emptyset) + \sum_{T \subseteq S, T \neq \emptyset} I_{\text{AND}}(T) + \sum_{T \cap S \neq \emptyset} I_{\text{OR}}(T) \right]$$

$$= \left[ \sum_{T \subseteq (S \cup \{i\}), T \neq \emptyset} I_{\text{AND}}(T) - \sum_{T \subseteq S, T \neq \emptyset} I_{\text{AND}}(T) \right] + \left[ \sum_{T \cap (S \cup \{i\}) \neq \emptyset} I_{\text{OR}}(T) - \sum_{T \cap S \neq \emptyset} I_{\text{OR}}(T) \right]$$

$$= \underbrace{\sum_{T \subseteq S} I_{\text{AND}}(T \cup \{i\})}_{\mathcal{A}} + \underbrace{\sum_{T \cap S = \emptyset} I_{\text{OR}}(T \cup \{i\})}_{\mathcal{B}}$$

In this way, we can decompose the Shapley value $\phi(i)$ into two terms, *i.e.*, $\phi(i) = \mathbb{E}_{S \subseteq N \setminus \{i\}}[\mathcal{X} + \mathcal{Y}] = \mathbb{E}_{S \subseteq N \setminus \{i\}}[\mathcal{X}] + \mathbb{E}_{S \subseteq N \setminus \{i\}}[\mathcal{Y}]$. Next, we first analyze the sum of AND interactions $\mathbb{E}_{S \subseteq N \setminus \{i\}}[\mathcal{X}]$, and then analyze the sum of OR interactions $\mathbb{E}_{S \subseteq N \setminus \{i\}}[\mathcal{Y}]$.

$$\mathbb{E}_{S \subseteq N \setminus \{i\}}[\mathcal{X}]$$

$$= \mathbb{E}_{S \subseteq N \setminus \{i\}} \sum_{T \subseteq S} I_{\text{AND}}(T \cup \{i\})$$

$$= \frac{1}{n} \sum_{m=0}^{n-1} \frac{1}{\binom{n-1}{m}} \sum_{\substack{S \subseteq N \setminus \{i\}, \\ |S| = m}} \sum_{T \subseteq S} I_{\text{AND}}(T \cup \{i\})$$

$$= \frac{1}{n} \sum_{T \subseteq N \setminus \{i\}} \sum_{m=0}^{n-1} \frac{1}{\binom{n-1}{m}} \sum_{\substack{S \supseteq T, \\ S \subseteq N \setminus \{i\}, \\ |S| = m}} I_{\text{AND}}(T \cup \{i\})$$

$$= \frac{1}{n} \sum_{T \subseteq N \setminus \{i\}} \sum_{m=|T|}^{n-1} \frac{1}{\binom{n-1}{m}} \sum_{\substack{S \supseteq T, \\ S \subseteq N \setminus \{i\}, \\ |S| = m}} I_{\text{AND}}(T \cup \{i\})$$

$$= \frac{1}{n} \sum_{T \subseteq N \setminus \{i\}} \sum_{m=|T|}^{n-1} \frac{1}{\binom{n-1}{m}} \binom{n-1-|T|}{m-|T|} I_{\text{AND}}(T \cup \{i\})$$

$$= \frac{1}{n} \sum_{T \subseteq N \setminus \{i\}} \underbrace{\sum_{k=0}^{n-1-|T|} \frac{1}{\binom{n-1}{|T|+k}} \binom{n-1-|T|}{k}}_{\alpha_T} I_{\text{AND}}(T \cup \{i\})$$

$$= \sum_{T \subseteq N \setminus \{i\}} \frac{1}{|T|+1} I_{\text{AND}}(T \cup \{i\})$$

$$= \sum_{S \subseteq N, i \in S} \frac{1}{|S|} I_{\text{AND}}(S) \quad /\!/ \text{ Let } S = T \cup \{i\}.$$

Then, for the sum of OR interactions, we have

$$\mathbb{E}_{S \subseteq N \setminus \{i\}}[\mathcal{B}]$$

$$= \mathbb{E}_{S \subseteq N \setminus \{i\}} \sum_{T \cap S \neq \emptyset} I_{\mathrm{OR}}(T \cup \{i\})$$

$$= \frac{1}{n} \sum_{m=0}^{n-1} \frac{1}{\binom{n-1}{m}} \sum_{\substack{S \subseteq N \setminus \{i\}, \, T \cap S \neq \emptyset \\ |S|=m}} \sum I_{\mathrm{OR}}(T \cup \{i\})$$

$$= \frac{1}{n} \sum_{T \subseteq N \setminus \{i\}} \sum_{m=0}^{n-1} \frac{1}{\binom{n-1}{m}} \sum_{\substack{S \cap T \neq \emptyset, \\ S \subseteq N \setminus \{i\}, \\ |S|=m}} I_{\mathrm{OR}}(T \cup \{i\})$$

$$= \frac{1}{n} \sum_{T \subseteq N \setminus \{i\}} \sum_{m=0}^{n-1} \frac{1}{\binom{n-1}{m}} \sum_{\substack{S \subseteq N \setminus \{i\} \setminus T, \\ |S|=m}} I_{\mathrm{OR}}(T \cup \{i\})$$

$$= \frac{1}{n} \sum_{T \subseteq N \setminus \{i\}} \sum_{m=0}^{n-1-|T|} \frac{1}{\binom{n-1}{m}} \sum_{\substack{S \subseteq N \setminus \{i\} \setminus T, \\ |S|=m}} I_{\mathrm{OR}}(T \cup \{i\}) \quad \text{// Since } S \subseteq N \setminus \{i\} \setminus T, |S| \leq n-1-|T|.$$

$$= \frac{1}{n} \sum_{T \subseteq N \setminus \{i\}} \sum_{m=0}^{n-1-|T|} \frac{1}{\binom{n-1}{m}} \binom{n-1-|T|}{m} I_{\mathrm{OR}}(T \cup \{i\})$$

$$= \frac{1}{n} \sum_{T \subseteq N \setminus \{i\}} \sum_{k=0}^{n-1-|T|} \frac{1}{\binom{n-1}{n-1-|T|-k}} \binom{n-1-|T|}{n-1-|T|-k} I_{\mathrm{OR}}(T \cup \{i\}) \quad \text{// Let } k = n-1-|T|-m.$$

$$= \frac{1}{n} \sum_{T \subseteq N \setminus \{i\}} \underbrace{\sum_{k=0}^{n-1-|T|} \frac{1}{\binom{n-1}{|T|+k}} \binom{n-1-|T|}{k}}_{\alpha_T} I_{\mathrm{OR}}(T \cup \{i\})$$

$$= \frac{1}{n} \sum_{T \subseteq N \setminus \{i\}} \frac{n}{|T|+1} I_{\mathrm{OR}}(T \cup \{i\})$$

$$= \sum_{T \subseteq N \setminus \{i\}} \frac{1}{|T|+1} I_{\mathrm{OR}}(T \cup \{i\})$$

$$= \sum_{S \subseteq N, i \in S} \frac{1}{|S|} I_{\mathrm{OR}}(S) \quad \text{// Let } S = T \cup \{i\}.$$

Therefore, $\phi(i) = \sum_{S \subseteq N \setminus \{i\}}[\mathcal{X}] + \sum_{S \subseteq N \setminus \{i\}}[\mathcal{Y}] = \sum_{S \subseteq N, i \in S} \frac{1}{|S|} I_{\mathrm{AND}}(S) + \sum_{S \subseteq N, i \in S} \frac{1}{|S|} I_{\mathrm{OR}}(S)$.

## J   SOLVING SATURATION PROBLEM

We prove that the emergence of high-order interactions is caused by the value shift (or called the *saturation problem*) of the value network, *i.e.*, most training data for the value network are usually biased/shifted to states with similar numbers of white stones and black stones, because in real games, the board always contains similar numbers of white stones and black stones. We use $\Delta n(T) = n_{\mathrm{white}}(T) - n_{\mathrm{black}}(T) \in \{-\frac{n}{2}, -\frac{n}{2}+1, ..., \frac{n}{2}\}$ to measure the unbalance level of the masked state $\boldsymbol{x}_T$, where $n_{\mathrm{white}}(T)$ and $n_{\mathrm{black}}(T)$ denote the number of white stones and that of black stones on $\boldsymbol{x}_T$, respectively. As Figure 7 shows, we compute the average advantage score $A_k =$

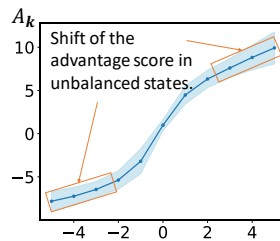

Figure 7: Value shifting over different orders. We show the average advantage score $A_k = \mathbb{E}_{\boldsymbol{x}}\mathbb{E}_{T \subseteq N:\Delta n(T)=k}\log(\frac{p_{\text{white}}(\boldsymbol{x}_T)}{1-p_{\text{white}}(\boldsymbol{x}_T)})$ over all masked states $\boldsymbol{x}_T$ with the same unbalance level $k$. The average advantage score $A_k$ is saturated when $|k|$ is large.

$\mathbb{E}_{\boldsymbol{x}}\mathbb{E}_{T \subseteq N:\Delta n(T)=k}\log(\frac{p_{\text{white}}(\boldsymbol{x}_T)}{1-p_{\text{white}}(\boldsymbol{x}_T)})$ over all masked states $\boldsymbol{x}_T$ with the same unbalance level $k \in \{-\frac{n}{2}, -\frac{n}{2}+1, ..., \frac{n}{2}\}$. The average advantage score $A_k$ is not roughly linear with $k$, but is saturated when $|k|$ is large.

Therefore, we propose to revise the advantage score $v(\boldsymbol{x}_T)$ in Equation (1) as $u(\boldsymbol{x}_T) = v(\boldsymbol{x}_T) - a_k$ to remove the value shift and thereby weaken high-order interactions. Given a masked state $\boldsymbol{x}_T$, we compute its unbalance level $k = \Delta n(T) = n_{\text{white}}(T) - n_{\text{black}}(T) \in \{-\frac{n}{2}, -\frac{n}{2}+1, ..., \frac{n}{2}\}$. $a_k$ is initialized as the average advantage score $A_k$.

Theorem 3 actually provides a family of AND-OR interactions *w.r.t.* the constraint $\forall T \subseteq N, u_T^{\text{AND}} + u_T^{\text{OR}} = u(\boldsymbol{x}_T)$, which all satisfy the universal matching property. Therefore, we parameterize the AND-OR interactions by setting $u_T^{\text{AND}} = 0.5(u(\boldsymbol{x}_T) + q_T) + p_T$ and $u_T^{\text{OR}} = 0.5(u(\boldsymbol{x}_T) + q_T) - p_T$ with a set of learnable parameters $\{p_T|T \subseteq N\}$. Thus, we extend the loss function in Equation (9), and use the following LASSO-like loss to learn the parameters $\boldsymbol{a} = \{a_{-\frac{n}{2}}, a_{-\frac{n}{2}+1}, ..., a_{\frac{n}{2}}\}, \{p_T|T \subseteq N\}, \{q_T|T \subseteq N\}$, and obtain the sparsest AND-OR interactions.

$$\min_{\boldsymbol{a},\{p_T\},\{q_T\}} \quad \|\boldsymbol{I}_{\text{AND}}(\boldsymbol{a}, \{p_T\}, \{q_T\})\|_1 + \|\boldsymbol{I}_{\text{OR}}(\boldsymbol{a}, \{p_T\}, \{q_T\})\|_1 \tag{15}$$

where $\|\cdot\|_1$ represents L-1 norm. $\boldsymbol{I}_{\text{AND}}(\boldsymbol{a}, \{p_T\}, \{q_T\}) = \text{vec}(\{ I_S^{\text{AND}} : S \subseteq N\}) \in \mathbb{R}^{2^n}$ and $\boldsymbol{I}_{\text{OR}}(\boldsymbol{a}, \{p_T\}, \{q_T\}) = \text{vec}(\{I_S^{\text{OR}} : S \subseteq N\}) \in \mathbb{R}^{2^n}$ denote all AND interactions and all OR interactions, respectively. $\boldsymbol{I}_{\text{AND}}(\boldsymbol{a}, \{p_T\}, \{q_T\})$ and $\boldsymbol{I}_{\text{OR}}(\boldsymbol{a}, \{p_T\}, \{q_T\})$ are parameterized by $\boldsymbol{a}, \{p_T\}, \{q_T\}$ according to Theorem 3.

In this way, given the learned parameters $\{u_T^{\text{AND}} : T \subseteq N\}$ and $\{u_T^{\text{OR}} : T \subseteq N\}$, numerical effects of all AND-OR interactions $\{I_S^{\text{AND}} : S \subseteq N\}$ and $\{I_S^{\text{OR}} : S \subseteq N\}$ can be directly computed by Theorem 3.

# K THE REASON WHY THE SATURATION PROBLEM CAUSES HIGH-ORDER INTERACTIONS

Let $g \in R$ and $h \in R$ denote the first derivative and second derivative of the curve of $A_k \overset{\text{def}}{=} \mathbb{E}_{\boldsymbol{x}}\mathbb{E}_{T \subseteq N:\Delta n=k}\log(\frac{p_{\text{white}}(\boldsymbol{x}_T)}{1-p_{\text{white}}(\boldsymbol{x}_T)})$ *w.r.t.* the $k$ value ($k \in \{-\frac{n}{2}, -\frac{n}{2}+1, ..., \frac{n}{2}\}$). Then, we can roughly consider that $A_k = A_0 + g \cdot k + \frac{h}{2} \cdot k^2$. Let us consider an interaction $S$ between $m$ stones, including $m_{\text{white}}$ white stones and $m_{\text{black}}$ black stones. The unbalance level of the masked board state $\boldsymbol{x}_S$ is $\Delta n = m_{\text{white}} - m_{\text{black}} = k^*$. If we only use AND interactions to explain the output of the value network, then we obtain the following equation.

$$v_m \overset{\text{def}}{=} \mathbb{E}_{T \subseteq S:|T|=m}[v(\boldsymbol{x}_T)] \approx A_{k^*}$$

$$v_{m'} \overset{\text{def}}{=} \mathbb{E}_{T \subseteq S:|T|=m'}[v(\boldsymbol{x}_T)]$$

$$\approx \frac{A_{k^*-((m-m'))}}{\binom{m_{\text{white}}}{m-m'}} + \frac{A_{k^*-((m-m'-1))}}{\binom{m_{\text{white}}}{m-m'-1}} + ... + \frac{A_{k^*+((m-m'-1))}}{\binom{m_{\text{black}}}{m-m'-1}} + \frac{A_{k^*+((m-m'))}}{\binom{m_{\text{black}}}{m-m'}} \tag{16}$$

$$v_0 \overset{\text{def}}{=} \mathbb{E}_{T \subseteq S:|T|=0}[v(\boldsymbol{x}_T)] \approx A_0$$

Note that $v_m, v_{m'}$ and $v_0$ are non-linear functions. The function $v_{m'}$ can be rewritten by following Taylor series expansion at the baseline point $m' = 0$ as follows.

$$v_{m'} = \mathbb{E}_{T \subseteq S:|T|=m'}[v(\boldsymbol{x}_T)] = v_0 + g_v \cdot m' + \frac{h_v}{2} \cdot m'^2 \tag{17}$$

where $g_v \in R$ and $h_v \in R$ denote the first derivative and second derivative of the curve of $v_{m'}$ w.r.t. the $m'$ value. In this way, the effect $I(S)$ of the interaction $S$ can be reformulated as follows.

$$
\begin{aligned}
I_S^{\text{AND}} &= \sum_{T \subseteq S} (-1)^{|S|-|T|} v(\boldsymbol{x}_T) \\
&\approx \binom{m}{0} v_m - \binom{m}{1} \cdot v_{m-1} + \binom{m}{2} \cdot v_{m-2} - \binom{m}{3} \cdot v_{m-3} + \binom{m}{4} \cdot v_{m-4} - ...
\end{aligned}
\tag{18}
$$

According to Equation (17), each component $v_{m'}$ of $I_S^{\text{AND}}$ consists of a term $\frac{h_v}{2} \cdot m'^2$. However, the term $\frac{h_v}{2} \cdot m'^2$ contained in $v_{m'}$ cannot cancel out with each other. Therefore, the interaction effect $I_S^{\text{AND}}$ will increase with the order of the interaction $S$.

## L  THE COMPOSITIONS OF HUMAN GO PLAYERS

We have collaborated with 5 professional human Go players to compare interactions/coalitions encoded by the value network with the human player's QiGan of the Go game. Some discovered shape patterns well fit human understanding, but other discovered patterns are beyond the common understanding of the game, and sometimes even conflict with traditional tactics of the Go game.

The following table shows the number of interactions of each order that are recognized as meaningful patterns by professional human players. Different players have identified almost the same list of shape patterns.

Interactions of the 1st order: All players consider that they are too simple to be analyzed.

Interactions of the 2nd order: about 62% of salient interactions are recognized.

Interactions of the 3rd order: about 23% of salient interactions are recognized.

Interactions of the 4th order: about 11% of salient interactions are recognized.

Interactions of the 5th order: about 5.8% of salient interactions are recognized.

Interactions of more than the 6th order: no interactions are recognized.

## M  THEOREMS AND PROPERTIES OF THE ATTRIBUTION METHOD IN EQUATION (11).

The coalition attribution satisfies the following desirable properties.

• **Symmetry property:** If the input variable $i \in N$ and the input variable $j \in N$ cooperate with other input variables in $S \subseteq N \setminus \{i,j\}$ in the same way, i.e. $\forall S \subseteq N \setminus \{i,j\}, v(S \cup \{i\}) = v(S \cup \{j\})$, then the coalition formed by $S \cup \{i\}$ and the coalition formed by $S \cup \{j\}$ have the same attribution, i.e., $\forall S \subseteq N \setminus \{i,j\}, \varphi(S \cup \{i\}) = \varphi(S \cup \{j\})$.

• **Additivity property:** If the output score of the model $v$ can be represented as the sum of the output score of the model $v_1$ and the output score of the model $v_2$, i.e. $\forall S \subseteq N, v(S) = v_1(S) + v_2(S)$, then the attribution of any coalition $S$ on the model $v$ can also be represented as the sum of the attribution of $S$ on the model $v_1$ and that on the model $v_2$, i.e. $\forall S \subseteq N, \varphi_v(S) = \varphi_{v_1}(S) + \varphi_{v_2}(S)$.

• **Dummy property:** If a coalition $S$ is a dummy coalition, i.e. $\forall i \in S, \forall T \subseteq N \setminus \{i\}, v(T \cup \{i\}) = v(T)$, then the coalition $S$ has no attribution on the model output, i.e. $\varphi(S) = 0$.

• **Efficiency property:** For any coalition $S$, the model output can be decomposed into the attribution of the coalition $S$ and the attribution of each input variable in $N \setminus S$ and the utilities of the interactions covering partial variables in $S$, i.e., $\forall S \subseteq N, v(N) - v(\emptyset) = \varphi(S) + \sum_{i \in N \setminus S} \varphi(i) + \sum_{T \subseteq N, T \cap S \neq \emptyset, T \cap S \neq S} \frac{|T \cap S|}{|T|} \left[ I_T^{\text{AND}} + I_T^{\text{OR}} \right]$

The following Equation (19) explains $\varphi(T) - \sum_{i \in T} \phi(i)$, i.e., the difference between the coalition's attribution $\varphi(T)$ and the sum of Shapley values $\phi(i)$ for all input variables $i$ in $T$, where the input variable $i$'s Shapley value (defined as $\phi(i) = \sum_{S \subseteq N \setminus \{i\}} \frac{|S|!(n-|S|-1)!}{n!} [v(\boldsymbol{x}_{S \cup i}) - v(\boldsymbol{x}_S)]$) can also be proved/explained as $\phi(i) = \sum_{S \ni i} \frac{1}{|S|} (I_S^{\text{AND}} + I_S^{\text{OR}})$. The difference comes from those interactions

that only contain partial variables in $T$, not all variables in $T$.

$$\varphi(T) - \sum_{i \in T} \phi(i) = \sum_{\substack{S \cap T \neq \emptyset, \\ S \cap T \neq T}} \frac{|S \cap T|}{|S|} (I_S^{\text{AND}} + I_S^{\text{OR}}) \tag{19}$$

And we try to use Corollary 7 and Equation (19) to explain the conflict between the Shapley value of input variables and the attribution of the coalition as follows.

**Corollary 7.** *If $\forall T \subseteq N, T \ni i, T \not\supseteq S, I_T^{\text{AND}} = I_T^{\text{OR}} = 0$, then $\phi(i) = \frac{1}{|S|}\varphi(S)$*

Corollary 7 shows that if a set $S$ of input variables is always memorized by the DNN as a coalition, and the DNN does not encode any interactions between a set $T$ of input variables, where $T$ only contains partial variables in $S$, *i.e.*, $\forall T \subseteq N, T \cap S \neq S, T \cap S \neq \emptyset, I_T^{\text{AND}} = I_T^{\text{OR}} = 0$, then the attribution $\varphi(S)$ of the coalition $S$ can be fully determined by the sum of the Shapley value $\phi(i)$ of all input variables in $S$. Otherwise, if the DNN encodes interactions between a set $T$ of input variables, where $T$ contains just partial but not all variables in $S$, then Equation (19) shows the conflict between individual variables' attributions and the coalition $S$'s attribution come from interactions containing just partial but not all variables in $S$.

# N  INCONSISTENCY PROBLEM IN ATTRIBUTION

We use the following example to introduce the inconsistency problem. We can simply consider a coalition $T$ (*e.g.*, $T = \{1, 2, 3\}$) of input variables as a singleton variable $[T]$, then we have a total of $n - 2$ input variables in $N' = \{[T], 4, 5, ..., n\}$. Let $\varphi([T])$ denote the attribution of $[T]$ computed on the new partition $N'$ of the $n - 2$ variables. Alternatively, we can also consider $x_1$, $x_2$, $x_3$ as three individual variables, and compute their attributions $\varphi(1)$, $\varphi(2)$, $\varphi(3)$ given the original partition of input variables $N = \{1, 2, ..., n\}$. However, for most attribution methods, $\varphi([T]) \neq \varphi(1) + \varphi(2) + \varphi(3)$. This is the inconsistency problem of attributions.

# O  EXPERIMENTAL DETAILS

## O.1  THE COMPUTER RESOURCES

Our experimental are conducted on a server, which is equipped with a CPU that has 4 cores and 16 threads, 126GB of RAM, and an SSD with 960GB capacity. The server also includes two NVIDIA GeForce RTX Titan GPUs.

## O.2  SETTINGS FOR THE GENERATION OF ONE BOARD CONFIGURATION

We use pre-trained networks published on https://github.com/lightvector/KataGo. We set the board size as 19x19, by letting the KataGo play games against itself, *i.e.*, letting the KataGo take turns to play the move of black stones and play the move of white stones, we can generate a board state.

## O.3  SETTINGS FOR THE EXTRACTION OF INTERACTIONS IN SECTION 2.2

The learning rate for the learnable vector $\boldsymbol{p}, \boldsymbol{q}$ exponentially decays from 1e-6 to 1e-7. In particular, each element $a_k$ in the vector $\boldsymbol{a}$ has different initial learning rates. Specifically, the learning rate of $a_k$ decayed from $\frac{1}{\binom{|k|}{\frac{n}{2}}} \cdot 1e - 6$ to $\frac{1}{\binom{|k|}{\frac{n}{2}}} \cdot 1e - 7$.

The threshold $\eta$ is a small scalar to bound unavoidable noises $\boldsymbol{q}$ in the network output, which is set to be $\eta = 0.38$ in experiments. Specifically, the threshold $\eta$ is computed by setting to be 0.01 time of the average strength of the top-1% most salient interaction. We compute all AND interactions $\{I_S^{\text{AND}} : S \subseteq N\}$ by setting $u_T^{\text{AND}}$ in Theorem 3 as $v(\boldsymbol{x}_T)$, and compute all OR interactions $\{I_S^{\text{OR}} : S \subseteq N\}$ by setting $u_T^{\text{OR}}$ in Theorem 3 as $v(\boldsymbol{x}_T)$. Then, all AND interactions $\{I_S^{\text{AND}} : S \subseteq N\}$ and all OR interactions $\{I_S^{\text{OR}} : S \subseteq N\}$ are arranged in descending order of their interaction strength.

# P    MORE EXPERIMENTAL RESULTS

## P.1    TRANSFERABILITY OF INTERACTIONS BETWEEN STONES THROUGH DIFFERENT BOARDS.

We conduct experiments to show the transferability of interactions between stones through different boards. As Figure 8 shows, the explained stones in game board states 1, 2, 3 are the same, but the contextual stones are different. We computed interactions between the same set of stones on different boards (with different contextual stones). Figure 8 shows that the same interaction exhibited similar effects on different boards. For example, for Game 1, $\varphi(\{1,2,9\}) = 1.58$, for Game 2, $\varphi(\{1,2,9\}) = 1.49$, and for Game 3, $\varphi(\{1,2,9\} = 1.50$. It means that the shape patterns found on one board can be transferred to another board.

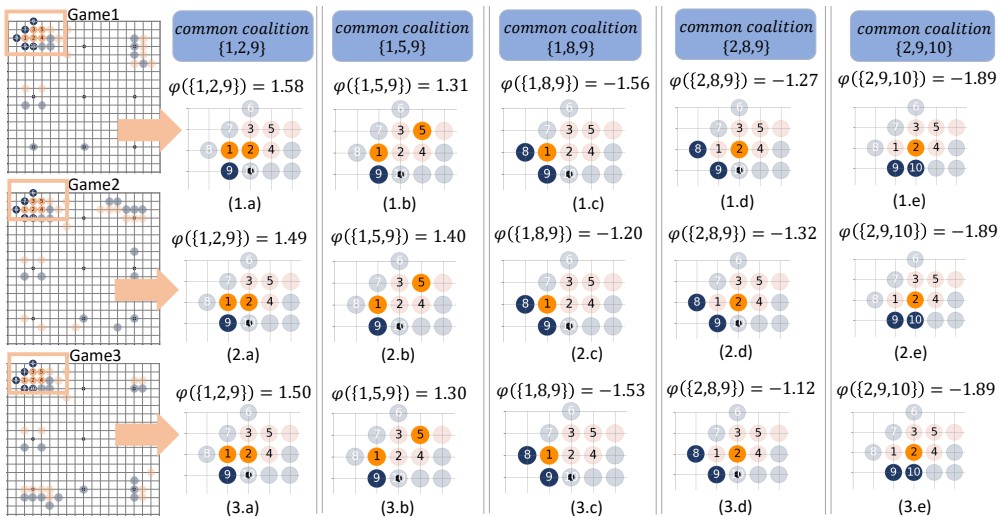

Figure 8: Transferability of interactions across different boards (with different contextual stones). Shape patterns extracted from one board can also explain the network outputs on other boards, *i.e.*, the same interaction contributes similar interaction effects to the advantage scores of different boards.

## P.2    GENERALIZATION POWER OF INTERACTIONS OF DIFFERENT ORDERS

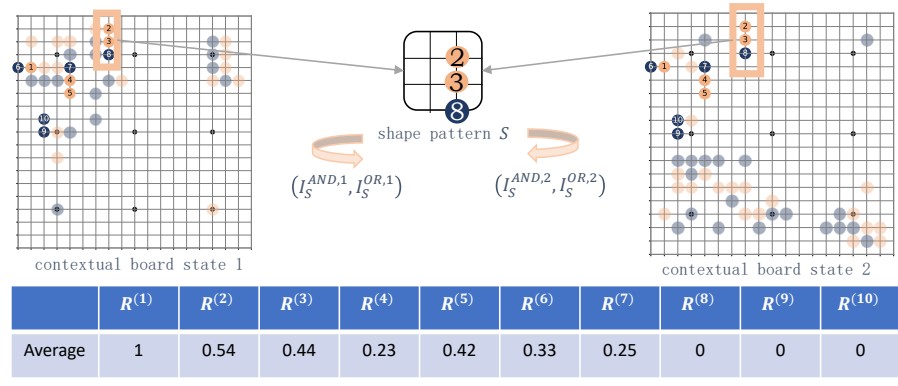

| | $R^{(1)}$ | $R^{(2)}$ | $R^{(3)}$ | $R^{(4)}$ | $R^{(5)}$ | $R^{(6)}$ | $R^{(7)}$ | $R^{(8)}$ | $R^{(9)}$ | $R^{(10)}$ |
|---|---|---|---|---|---|---|---|---|---|---|
| Average | 1 | 0.54 | 0.44 | 0.23 | 0.42 | 0.33 | 0.25 | 0 | 0 | 0 |

Figure 9: Generalization rate of interactions of different orders.

We have conducted experiments to evaluate the robustness/generalization power of shape patterns of each $m$-order in different contexts. Figure 9 reports the ratio of $m$-order interactions that can be generalized through different contextual board states, denoted by $R^{(m)}$. Specifically, we put the same shape pattern $S$ in two different contextual board states. We report the average generalization rate

$R^{(m)}$ of $m$-order interactions, which is averaged over different board states.

$$R^{(m)} = \frac{\text{\# of m-order generalizable interactions}}{\text{\# of all salient m-order interactions}}$$

where $\text{\# of all salient m-order interactions} = \sum_{S \subseteq N, |S|=m} \mathbb{1}(|I_S^{\text{AND},1}| > \xi) + \mathbb{1}(|I_S^{\text{OR},1}| > \xi)$, and $\text{\# of m-order generalizable interactions} = \sum_{S \subseteq N, |S|=m} \mathbb{1}(|I_S^{\text{AND},1}| > \xi \text{ AND } |I_S^{\text{AND},2}| > \xi) + \mathbb{1}(|I_S^{\text{OR},1}| > \xi \text{ AND } |I_S^{\text{OR},2}| > \xi)$. $(I_S^{\text{AND},1}, I_S^{\text{OR},1})$ and $(I_S^{\text{AND},2}, I_S^{\text{OR},2})$ denote the AND-OR interaction extracted under the first context and those under the second context, respectively. We find that high-order interactions ($m \geq 6$) exhibit lower generalization power.

### P.3 MORE SHAPE PATTERNS EXTRACTED FROM THE VALUE NETWORK FOR THE GAME OF GO.

We show more shape patterns extracted from the value network for the game of Go.

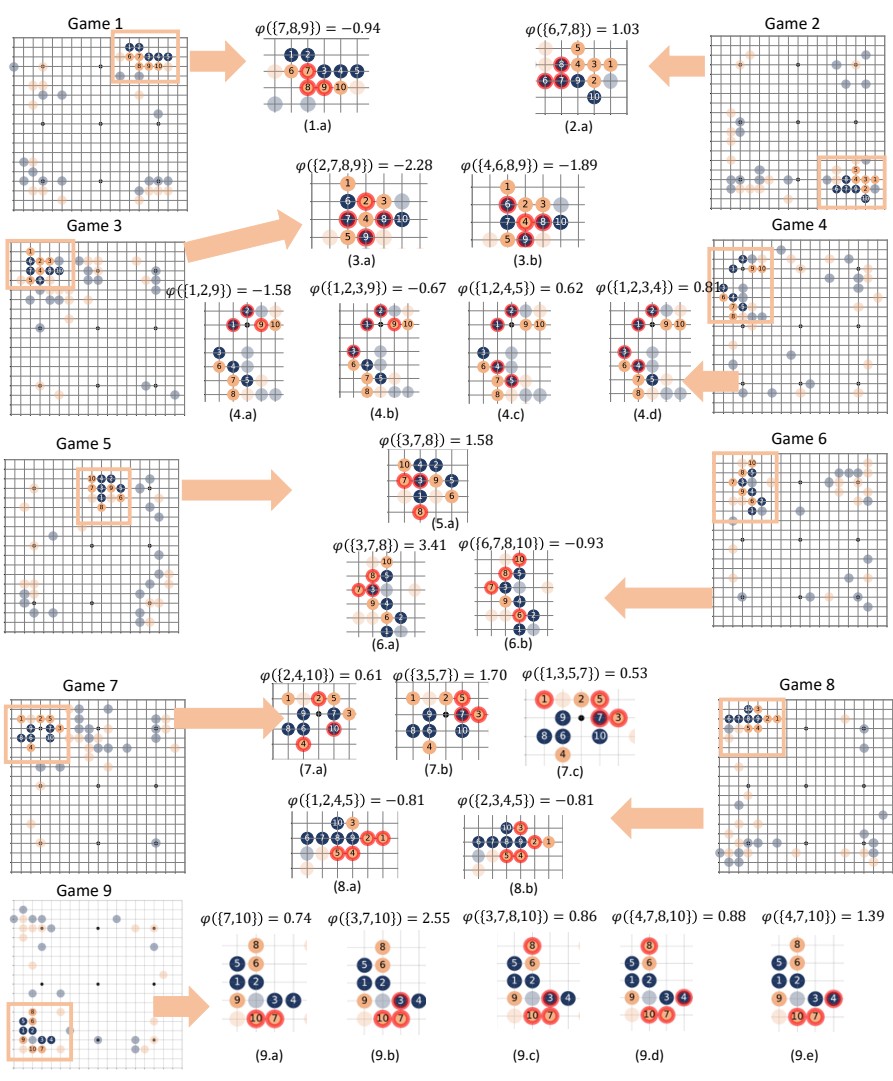

Figure 10: More experimental results for the estimated attributions of different coalitions (shape patterns). Stones in the coalition are highlighted by red circles.

**For Game 1** in Figure 10 (1.a), Go players are confused about why the coalition $\{7, 8, 9\}$ is advantageous for black stones.

**For Game 2** in Figure 10 (2.a), Go players cannot figure out why the coalition $\{6, 7, 8\}$ is advantageous for white stones.

**For Game 3** in Figure 10 (3.a-3.b), Go players consider that the black stones $x_6$, $x_7$ are caught, and the white stones are an advantage. However, the value network thinks that the coalition $\{2, 7, 8, 9\}$ and the coalition $\{4, 6, 8, 9\}$ are advantageous for black stones. Go players are confused about that.

**For Game 4** in Figure 10 (4.a-4.d), $\varphi(\{1, 2, 9\}) < \varphi(\{1, 2, 3, 9\})$, which means that the black stone $x_3$ is a low-value move; Go players consider that the stone $x_3$ a valuable move.

**For Game 5** in Figure 10 (5.a), $\varphi(\{3, 7, 8\}) = 1.58$ means that the white stones are in advantage. Furthermore, this shape pattern is considered a classic shape among Go players, aligning with their strategic understanding of the game.

**For Game 6** in Figure 10 (6.a-6.b), $\varphi(\{6, 7, 8, 10\}) = -0.93$, it is confusing for Go players that the black stones hold an advantage in the shape pattern $\{6, 7, 8, 10\}$. While $\varphi(\{3, 7, 8\}) = 3.41$, the shape pattern $\{3, 7, 8\}$ is advantageous for white stones, the attribution score aligns with Go players' strategic understanding of the game. Because it is a typical formalized series of moves (Dingshi) among Go players, known as "corner regular form."

**For Game 7** in Figure 10 (7.a-7.c), first, $\varphi(\{2, 4, 10\}) = 0.61$ and $\varphi(\{3, 5, 7\}) = 1.70$, which means that white stones are in advantage in the shape pattern $\{2, 4, 10\}$ and $\{3, 5, 7\}$. Meanwhile, Go players consider that $\{2, 4, 10\}$ and $\{3, 5, 7\}$ are classic shape patterns, and the attribution scores of these two coalitions align with their understanding of the Go game. Second, $\varphi(\{2, 4, 10\}) = 1.70$ and $\varphi(\{1, 3, 5, 7\}) = 0.53$. The shape pattern $\{3, 5, 7\}$ is a "shoulder-hit" pattern; therefore, white stones are an advantage in this shape pattern. However, the white stone $x_1$ makes the opportunity for black stones to break the union of white stones; therefore, $\varphi(\{1, 3, 5, 7\}) < \varphi(\{3, 5, 7\})$.

**For Game 8** in Figure 10 (8.a-8.b), $\varphi(\{1, 2, 4, 5\}) = -0.81$ and $\varphi(\{2, 3, 4, 5\}) = -0.81$, which means that black stones hold an advantage in these two shape patterns. Because in these two cases, white stones form a combat configuration too slowly.

**For Game 9** in Figure 10 (9.a-9.e), (1) $\varphi(\{7, 10\}) = 0.74$ and $\varphi(\{3, 7, 10\}) = 2.55$. The combination of white stones $x_7$ and $x_10$ is not a good strategy. However, the move of black stone $x_3$ makes the combination of white stones $\{7, 10\}$ form a combat configuration. The shape pattern $\{3, 7, 10\}$ is a typical tactical pattern known as "shoulder-hit." Therefore, shape pattern $\{3, 7, 10\}$ is more advantageous for white stones than shape pattern $\{7, 10\}$. (2) $\varphi(\{7, 10\}) = 0.74$ and $(\{4, 7, 10\}) = 1.39$. Likewise, the move of black stone $x_4$ also makes the combination of white stones $\{7, 10\}$ form a combat configuration. The shape pattern $\{4, 7, 10\}$ is also a "shoulder-hit" pattern, and is more advantageous for white stones than the shape pattern $\{7, 10\}$. (3) However, the position of black stone $x_4$ in the pattern $\{4, 7, 10\}$ is superior to the black stone $x_3$ in the pattern $\{3, 7, 10\}$. Therefore, $\varphi(\{4, 7, 10\}) < \varphi(\{3, 7, 10\})$. (4) $\varphi(\{3, 7, 10\}) = 2.55$ and $\varphi(\{3, 7, 8, 10\}) = 0.86$. Although the shape pattern $\{3, 7, 10\}$ is a "shoulder-hit" pattern, due to the problematic placement of white stone $x_8$, black stones get the opportunity to split the white combinations $\{7, 8, 10\}$. Therefore, shape pattern $\{3, 7, 8, 10\}$ have a lower advantage score than shape pattern $\{3, 7, 10\}$. (5) $\varphi(\{4, 7, 10\}) = 1.39$ and $\varphi(\{4, 7, 8, 10\}) = 0.88$. Although the shape pattern $\{4, 7, 10\}$ is a "shoulder-hit" pattern, due to the problematic placement of white stone $x_8$, black stones get the opportunity to split the white combinations $\{7, 8, 10\}$. Therefore, shape pattern $\{4, 7, 8, 10\}$ have a lower advantage score than shape pattern $\{4, 7, 10\}$.

P.4 HOW TO UNDERSTAND THE INTERACTIONS BETWEEN STONES AND EMPTY POINTS?

The interaction with a vast empty region on the board is an interesting question. In mathematics, the vast empty region $S$ in the board can be represented as a special OR interaction. If any stone is placed on the region $S$, it will activate the OR interaction, which means the vast empty region has been destroyed. Only when no stone is placed on the region $S$, the silence of the OR interaction means the existence of a vast region. If the vast empty region has a positive effect on the advantage score, it is equivalent to letting the DNN encode an OR interaction with a negative effect.

## P.5 EXTENSION OF OUR METHOD TO EXPLAIN GOBANG GAME

We apply our method to another application, i.e., the Gobang game. Specifically, we analyze the value network in the open-source Gobang project, Katagomo. The Katagomo model is designed upon the KataGo project, and it also has a value network. We use the Katagomo model to generate a Gobang board with 20 stones. Then, we use our method to analyze shape patterns encoded by Katagomo's value network. Furthermore, we compute the attribution $\varphi(T)$ of each shape pattern encoded by the katagomo's value network. Figure 11 compares the strength of interactions of different orders. Given different Gobang game board states, our method in Equation (15) extracts weaker high-order interactions than the original method in Equation (9). Figure 12 visualizes the common coalitions selected from two gobang game states.

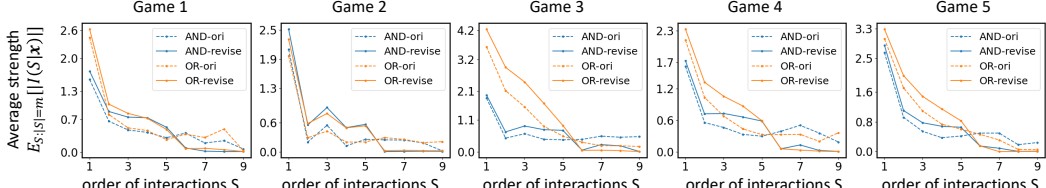

Figure 11: Average strength of effects for interactions of different orders. For different gobang games, our revised method extracts weaker high-order interactions than the original method.

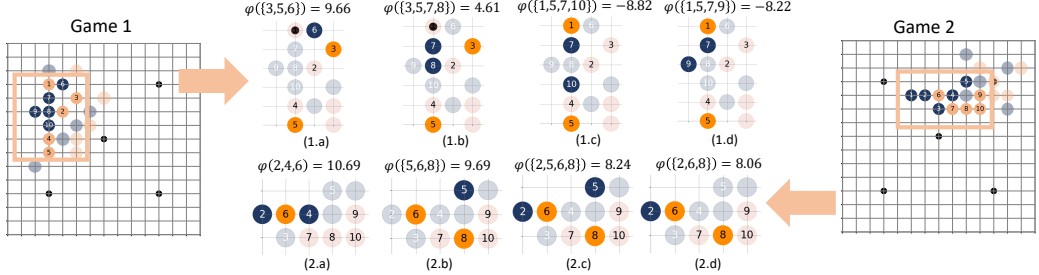

Figure 12: Estimated attributions of different coalitions (shape patterns) selected from two gobang game states.

## P.6 EXTENSION OF OUR METHOD TO DEBUG REPRESENTATION FLAWS HIDDEN IN A DNN

The second application of our method is to debug representation flaws hidden in a DNN. Besides letting people learn new interactions/concepts encoded by the DNNs, another typical utility of explaining interactions in a DNN is to debug representation flaws hidden in a DNN.

Figure 13 shows the interactions extracted from a DNN for pedestrian detection. Given an input image, we manually label image regions with salient attributions as input variables, and compute interactions between image regions. The visualization of the interactions enables people to check the correctness of interactions encoded by the DNN manually. Let us consider the explanation on the first input image as an example. We can analyze the representation quality of the DNN from the following three perspectives. (1) The interactions $I_{\text{AND}}(S = \{C, I\})$, $I_{\text{AND}}(S = \{D, I\})$, $I_{\text{AND}}(S = \{F, I\})$ and $I_{\text{AND}}(S = \{A, I\})$ between pedestrian patches and background patches may represent unreliable inference patterns. (2) High-order interactions, *e.g.*, $I_{\text{OR}}(S = \{A, C, D, E, F, G, H, I\})$ and $I_{\text{OR}}(S = \{A, C, D, F, H, I, J\})$, usually represent too complex inference patterns. Complex interactions usually have lower generalization power than simple interactions. (3) There are 29 positive interactions and 31 negative interactions extracted from an input image. The offsetting of positive and negative interactions is another problem. Adversarially robust neural networks usually encode more positive interactions and fewer negative interactions than normal neural networks.

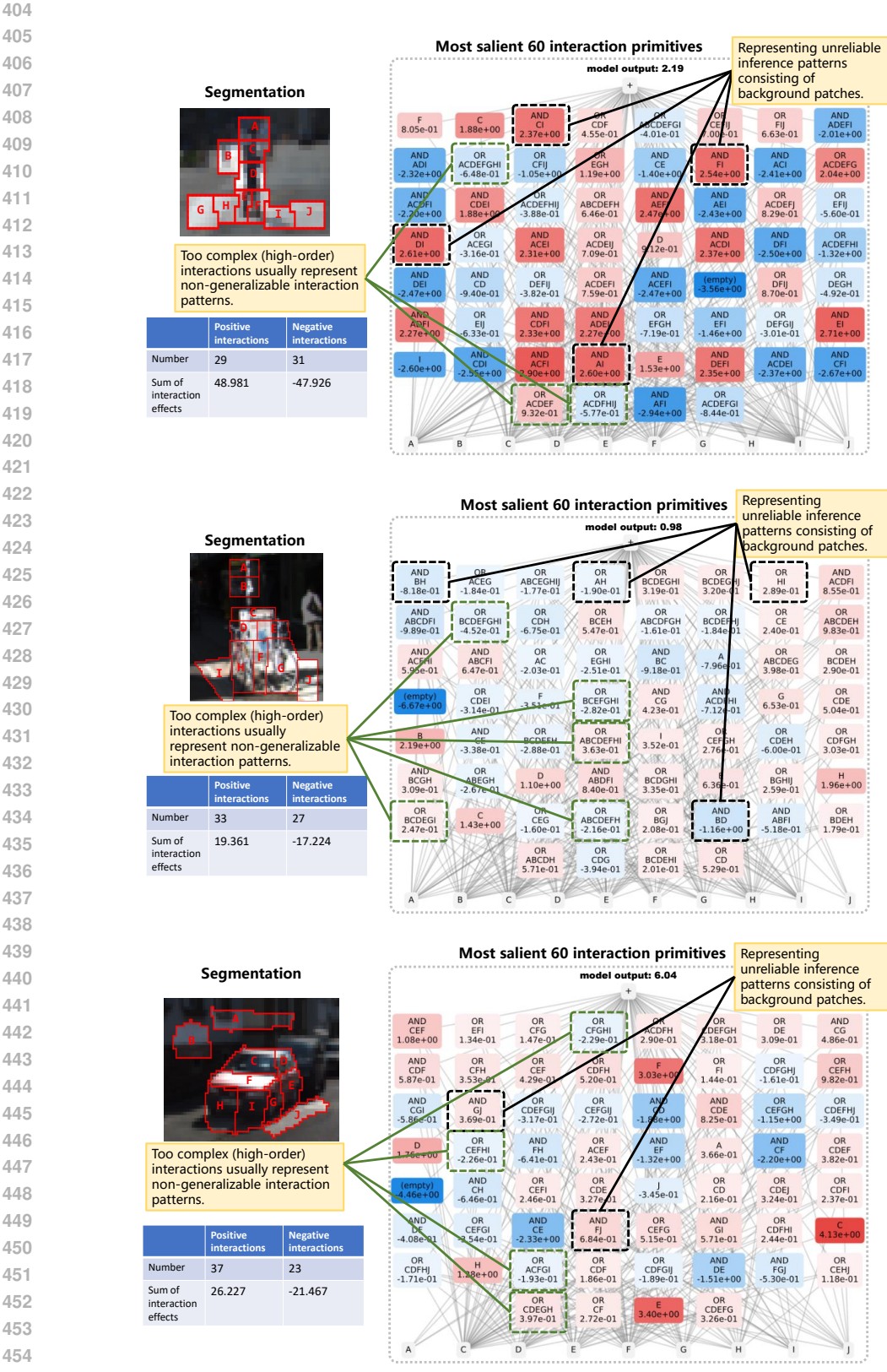

Figure 13: The interactions extracted from a DNN for pedestrian detection.

In addition, the problematic interactions (*e.g.*, interactions on background patches) reflect representation flaws of a DNN, because it is found by Li & Zhang (2023) that salient interactions are usually transferable across different samples. In other words, problematic interactions may affect the inference of a large number of samples.

## Q    THE USE OF LARGE LANGUAGE MODELS (LLMs)

In this study, LLMs were only used to correct grammatical errors and improve non-native expressions. **Research ideation did not involve the use of LLMs**. Specifically, the authors first wrote the original version of this manuscript in their native language without the assistance of LLMs. Subsequently, this version was divided into segments and provided to GPT-4.1 for translation and language polishing. Finally, the authors manually revised the English version themselves. The authors take full responsibility for the contents of this paper.

