# OpenReview forum: "Quantifying QiGan: What Shape Patterns Does a DNN Exploit in Go?"
_ICLR.cc/2026/Conference — ICLR 2026 Conference Withdrawn Submission_

### Official Review · Reviewer_M2SY · 2025-10-25

**Soundness:** 3
**Presentation:** 3
**Contribution:** 3
**Rating:** 6
**Confidence:** 3

**Summary:**

This paper investigates how deep neural networks (DNNs) that surpass human performance in Go encode interpretable knowledge. The authors introduce a framework to disentangle primitive “shape patterns” (QiGan) encoded in a Go value network (KataGo). They propose a novel AND–OR logical model that extends previous AND interaction frameworks to capture both conjunction and disjunction relationships among stones. The method theoretically guarantees faithfulness through a universal matching property and promotes sparsity to ensure concise explanations. Empirical results (Figure 2–6; Section 2.3–2.5) demonstrate that AND–OR interactions provide more compact, transferable, and human-understandable shape patterns than purely AND-based models. Collaborations with professional Go players confirm that some extracted patterns align with human intuition while others reveal new, previously unknown tactics.

**Strengths:**

1. The paper establishes a solid theoretical foundation for interpretability by formally proving the universal matching property of AND–OR interactions, ensuring that the proposed logical model can exactly reproduce the value network’s outputs.

---

2. The work meaningfully extends previous AND-only interaction frameworks by introducing OR relations, showing that disjunctive dependencies cannot be concisely represented using conjunctions alone.

---

3. Collaborations with professional Go players provide qualitative validation: some discovered coalitions align with human intuition, while others reveal previously unrecognized strategies, offering both interpretive alignment and discovery value.

**Weaknesses:**

1. The current evaluation analyzes only (n = 10) stones per board (Sec. 2.3), manually chosen by a human expert. This small and subjective subset introduces potential bias, as the representativeness of the selected stones is unverified. Moreover, the effect of this manual selection on the discovered patterns, sparsity, and interaction statistics is not quantified. The study also lacks evidence regarding scalability to full-board (361-position) analysis, leaving uncertainty about whether the approach can generalize beyond local subregions.

---

2. The interpretability validation primarily relies on qualitative feedback from a few professional Go players (Sec. 2.6). While these insights are valuable, the study does not include structured or quantitative assessment methods, such as inter-rater agreement, controlled blind testing, or statistical summaries of player judgments. As a result, the connection between the extracted coalitions and human reasoning remains anecdotal, and it is unclear to what extent the discovered patterns genuinely enhance human understanding.

---

3. Although the framework involves evaluating (2^n) masked states per position (Sec. 2.1), the paper provides no runtime, memory, or computational complexity analysis. The pseudocode in Appendix A outlines procedural steps but omits time-cost or scalability estimates. Without such information, it is difficult to judge whether the approach is computationally feasible for larger networks or higher board resolutions, which limits reproducibility and broader applicability.

**Questions:**

**1. Section 2.1:**

The paper requires evaluating outputs over all (2^n) masked board states (Eq. 2–4). While feasible for (n=10), this appears exponentially expensive for larger inputs. Could the authors clarify whether any sampling, pruning, or approximation strategy is used to reduce computational cost? How would this framework scale to full-board (361-stone) analysis, even approximately?

---

**2. Section 2.3:**

Only (n = 10) stones are analyzed per board, selected manually by a professional player. How sensitive are the extracted interactions and sparsity patterns to this manual selection? Have the authors tested random or automated subset selection to confirm robustness? Including such ablations could help assess whether the observed conciseness and fidelity trends (Fig. 2, Fig. 5b) are consistent across different subsets.

---

**3. Section 2.2:**

Theoretical justification (Theorem 2) shows that OR relations cannot be efficiently represented by AND-only models. Could the authors provide quantitative evidence demonstrating how OR interactions improve sparsity or fidelity in practice? For instance, how many salient terms are reduced when OR terms are introduced compared with AND-only baselines? Are there any cases where OR interactions negatively affect interpretability?

---

### Official Review · Reviewer_AaoL · 2025-10-27

**Soundness:** 3
**Presentation:** 3
**Contribution:** 3
**Rating:** 6
**Confidence:** 4

**Summary:**

In this paper, the authors propose a novel approach for extracting "QiGan" employed by pre-trained Go models during inference. This method is based on interactions, which encode logical relationships between stones. The authors extend the traditional AND interactions to OR interactions, propose methods to eliminate non-transferable higher-order interactions, and ultimately extract "common coalitions" to represent the "QiGan" encoded by the Go models. Through collaboration with professional Go players, the authors show that the extracted "QiGan" often aligns with human knowledge and can also provide novel insights previously unknown to humans.

**Strengths:**

1. The authors offer insights into pre-trained Go models from a relatively novel perspective, and their interpretability results have been validated by professional players.

2. The method proposed in the paper appears to be theoretically well-founded and is accompanied by relatively rigorous mathematical derivations.

3.The proposed method is highly generalizable. It can not only be used to interpret models applied in board games, but can also be extended to other domains such as computer vision. This approach could be further developed into a general framework for explaining any deep neural network.

**Weaknesses:**

1. To understand the "interactions" used in this paper, it is necessary to read and comprehend the lengthy and complex preliminaries. Moreover, the methods proposed in this paper are not easy to follow. Although the authors provide some intuitive insights when explaining the concepts, the paper remains quite challenging to read.

2. The authors did not specify whether this method could be used to guide improvements in Go models.

**Questions:**

At the end of Section 2.2, the author introduces a parameter {ϵ_T} to eliminate the output noise of the model v(⋅). However, the authors do not provide a theoretical analysis of how the parameter  {ϵ_T} might affect the proposed method. Could the authors consider supplementing this part with some theoretical discussion?

---

### Official Review · Reviewer_69AU · 2025-10-31

**Soundness:** 2
**Presentation:** 1
**Contribution:** 2
**Rating:** 2
**Confidence:** 3

**Summary:**

This paper proposes a method to disentangle the primitive shape patterns, encoded in a superhuman AI model for Go, to enhance the human knowledge about the game of GO. The approach aims to provide a concise, faithful, and verifiable explanation of the knowledge the AI uses for fast board-state assessment. The experiments show this method successfully uncovered many novel shape patterns that were not part of traditional human Go knowledge.

**Strengths:**

- The idea of extracting human understandable patterns from superhumans AI models is quite interesting and I think it is a promising research direction.

**Weaknesses:**

- The paper is very hard to follow, and I don't understand the motivation of the different approaches taken by the authors. Is there something more than just trying to explain a black-box model with a logic-based network? Considering this has been done for decades in the AI community, the only novelty I see is about trying to connect the explanation with the game of GO and then to GO players..
- Related work discussion is completely missing.
- The paper is almost unstructured, with section 2 spanning over the full paper. To make it more readable it should be completely rewritten.  In my opinion the paper cannot absolutely be accepted in this current form.
- Footnotes are wildly used throughout the whole paper making it even worse to be read.
- While the general idea of the paper is quite interesting, all the methodology here is very specific to the game of go, and then quite difficult to assess if the contribution of the paper is enough for a top-level conference like ICLR.

MINOR COMMENT AND TYPOS:
- The footnote on QiGan in the abstract is repeated. Also I'd avoid the usage of too many footnotes in general, and especially in the abstract.
- "e.g.chess" -> e.g. chess
- footnote numbered 13 appears after footnote 4.. and footnotes about terms like QiGan is repeated through all the paper... Cannot figure it out the reason for this practice..
- footnote 3 send to the video in supplemental material..

**Questions:**

1) What is a shape pattern, can the author provide an example? I only played GO once, and similarly the paper should be enjoyable even for readers that are not familiar with the game.
2) What does it meant exactly by "interaction"? Is just a latent property possessed by all or a sub-groups of stones? Formally this is just a subset of the stones that are on the board?

---

### Official Review · Reviewer_SPst · 2025-11-02

**Soundness:** 1
**Presentation:** 2
**Contribution:** 1
**Rating:** 2
**Confidence:** 2

**Summary:**

The paper tries to extract knowledge from the value network of a network trained for the Game of GO. I.e what are the stone patterns the networks uses to derive a value score (winning chances of white/black) for a given board state. Prior work tries to explain using AND operation between stones (e.g. if there is a white stone at locations A AND B AND C --> give score X to White). This paper proposes to also add OR interactions between stones.

It is claimed and prooved that this leads to fewer total interactions (i.e number of rules), which makes sense.

In practice, there are in total 50 game states, in each only 10 expert-selected stones of all is being studied (line 307).

There are some results showing how the total number of interaction reduces when OR interactions of this paper are introduced. Also some sample results on the patterns with different scores and how to interpret them (e.g. Fig 6) which I'm however not sure if they have any OR interactions.

**Strengths:**

A) I generally think the setup of trying to distill a Value Network of a challenging domain (Go game) to simple AND-OR interactions for discovering patterns is interesting (though this is not the contribution of the paper and is prior works’).

B) The figures are mostly easy to read. Except for Fig. 6, which I think should have the board games on one side, and the patterns on the other. But aside from this, it wasn’t hard to understand the figures.


C) The notation is very complete (and verbose as discussed in concern B). But the authors are thorough in explaining their theoretical claims.

D) Going beyond AND-only interaction makes sense to me, although I have a hard-time justifying the introduction of OR interactions (see concern C)

**Weaknesses:**

**(A) Writing the Introduction Section**

The current introduction heavily assumes the reader knows the Go Game, the black box nature of existing models, and how models like Alpha Go have beaten the human expert. This should be re-written from a more general introduction, down to the “case study” of the paper over the Go Game.

Something in lines of: 1. AI models have shown to out-perform human experts, yet understanding their inner workings is not clear. (followed by examples in protein folding, games, etc.) 2. In this paper we run a case study on the Challenging Go game, as one of the complex examples that the current models outperform humans.

It should motivate why we are looking at the Go Game and how complex it is, and why the case study is a good proxy for other tasks such as drug discovery etc. Without this, if the reader is not interested in the Go Game itself, they might skip the paper. This information is there, already in the introduction, but it keeps getting too detailed. For example, the footnote on AlphaGo not being accessible etc. absolutely does not belong to the first page! Similarly, the second paragraph which discusses the “value network”, “policy network”, or the Tree Search are all again too detailed for a non-RL-expert reader and belong to the Method/RelatedWork section.

Even as someone who is familiar with the game of Go, Alpha Go, and general RL concepts, I was lost reading the second page of the introduction.



**(B) Method Notation**
I was able to follow most of the notations, although I think it can be made much shorter and less verbose. In a nutshell, the method section describes the principles of sparsity in terms of number of interactions, faithfulness in terms of interactions being able to mimic the value networks over different inputs (a distillation in a sense), and the two types of AND (prior work) and OR interactions (this work) between individual pieces.

This however should be communicated more efficiently. For example, see Eq. 6. Basically all of this notation wants to say that an OR operation over M boolean variables, can be explained as a $2^m$ AND operation, covering all possible states. This is something that can be communicated simply in words, and 90% of the readers would agree, and others can be referred to the Appendix for a detailed proof (also a simple visualization with three variables would suffice). But instead there is the extremely detailed math notation (Eq 6) which may throw off the readers.

Similarly, for example in Eq 7, every time the set $S$ is mentioned, it is followed by $\neq \emptyset$. This should at most be mentioned once at the beginning of the method section, and not be repeated in every equation.

**(C) Method Implementation and Setup**
The method section goes in great detail over the defined interactions and its theorems. However, many key questions are not answered.

- How are the interactions (and their $I$ values) trained? Do we have a multi-layer logic network? I.e interactions defined over the results of previous interactions?


- Do we train our interaction scores $I$ per-game or do we have a collection of games and a dataset? i.e do we have different score values for the same interactions across games?
If we are learning the values of interactions **per-board**, I have a hard-time justifying the learned values would be any meaningful. Essentially we would be overfitting to a certain arrangement of 10 individual stones (line 307). This, as far as I understand, would not yield a general rulebook of patterns that the network uses.

- How complex is the case-study model (i.e. KataGo’s value network)? How good are the extracted logic interactions in approximating it? Essentially, what is the final performance on Eq. 8 (as it’s a learning process)?

- What are existing baselines? There is a ton of literature on pattern-mining, concept, and circuit discovery. Could the authors elaborate on why these do not apply here? Besides the AND-only prior work that this work builds on, what are other works trying to explain KataGo's knoweledge?


**(D) Method Motivation**

The main contribution of the paper is centered around having OR interaction, e.g. assigning a score if (Stone A OR Stone B OR Stone C) is present. I really don’t understand how this would be a useful interaction to find. It is shown in Fig 2, that the introduction of OR interaction limits the total number of interactions needed, which I get why is desirable. However, does this encode useful information? Essentially this assigns importance values to individual stones and is not really a “pattern”. What would make sense to me is having OR interaction over patterns rather than stones (i.e if AND_interaction_A is present OR AND_interaction_B is present). Could the authors further clarify this?

I tried looking for an example figure showing an interesting OR interaction in the paper, but couldn’t find any. The closest I got was Fig. 6, where the importance of different patterns are reported and compared, yet these are again all AND interactions as far as I understood.
I also looked at the video supplement, but the video had examples that were not Go-related, and explained the concepts in terms of LLMs and vision models.

**Questions:**

See concerns (C) and (D).

---

### Note · Authors · 2025-11-22

I have read and agree with the venue's withdrawal policy on behalf of myself and my co-authors.